# Nuclear pores safeguard the integrity of the nuclear envelope

Reiya Taniguchi ®[1,7], Clarisse Orniacki ®[2,8], Jan Philipp Kreysing ®[1,3], Vojtech Zila ®[4,9], Christian E. Zimmerli ®[1,10], Stefanie Böhm ®[1], Beata Turoňová ®[1], Hans-Georg Kräusslich ®[4,5], Valérie Doye ®[2] ✉ & Martin Beck ®[1,6] ✉

Nuclear pore complexes (NPCs) mediate nucleocytoplasmic exchange, which is essential for eukaryotes. Mutations in the central scaffolding components of NPCs are associated with genetic diseases, but how they manifest only in specific tissues remains unclear. This is exemplified in Nup133-deficient mouse embryonic stem cells, which grow normally during pluripotency, but differentiate poorly into neurons. Here, using an innovative in situ structural biology approach, we show that $Nup133^{-/-}$ mouse embryonic stem cells have heterogeneous NPCs with non-canonical symmetries and missing subunits. During neuronal differentiation, Nup133-deficient NPCs frequently disintegrate, resulting in abnormally large nuclear envelope openings. We propose that the elasticity of the NPC scaffold has a protective function for the nuclear envelope and that its perturbation becomes critical under conditions that impose an increased mechanical load onto nuclei.

The nucleus is the eukaryotic organelle that stores genetic material, and its secure maintenance is essential for cell survival. It is surrounded by the nuclear envelope, which consists of two lipid bilayers—the outer and inner membranes. Nuclear pore complexes (NPCs) are embedded in the nuclear envelope and regulate nucleocytoplasmic exchange. Structurally, the NPC has an eight-fold symmetric, cylindrical architecture, which can be subdivided into three rings stacked along the central axis: two outer rings (the cytoplasmic ring (CR) and nuclear ring (NR)) at their respective sides of the nuclear envelope, and the inner ring (IR) at the fusion point of the nuclear membranes[1,2]. Each ring is composed of specific subcomplexes formed by multiple protein components called nucleoporins (Nups). In mammalian NPCs, the Y-complex (Nup107–160 complex)[3–5] oligomerizes into two tandem rings[6] forming the scaffold of the CR and NR[7–10]. This is mediated by a head-to-tail contact between Nup133 of one Y-complex and Nup160 of the adjacent Y-complex[9,11]. Despite the structural importance of the Y-complex for NPC architecture, mutations in Y-complex Nups are known to only affect the development and function of specific tissues, such as kidney[12–14], ovary[15] or brain[16–18]. Moreover, certain Y-complex Nups, including Nup133, are dispensable in mouse embryonic stem (mES) cells[19–22], but their absence severely affects cell differentiation[19,21–23]. Such tissue- and cell type-specific phenotypes caused by scaffolding Nup genetic defects are difficult to conceive because they cannot be explained by global NPC misassembly or malfunction.

NPCs are known to be conformationally dynamic within cells. For instance, *Schizosaccharomyces pombe* NPCs reversibly constrict under hyper-osmotic stress, when nuclear shrinkage and nuclear envelope ruffling occur[24]. Similarly, human NPCs have a larger diameter in situ[25–27]

[1]Department of Molecular Sociology, Max Planck Institute of Biophysics, Frankfurt am Main, Germany. [2]Université Paris Cité, CNRS, Institut Jacques Monod, Paris, France. [3]IMPRS on Cellular Biophysics, Frankfurt am Main, Germany. [4]Department of Infectious Diseases, Virology, Heidelberg University, Heidelberg, Germany. [5]German Centre for Infection Research (DZIF), Partner Site Heidelberg, Heidelberg, Germany. [6]Institute of Biochemistry, Goethe University Frankfurt, Frankfurt am Main, Germany. [7]Present address: RIKEN Center for Integrative Medical Sciences, Tsurumi-ku, Yokohama, Japan. [8]Present address: The Neuro – Montreal Neurological Institute and Hospital, McGill University, Montreal, Quebec, Canada. [9]Present address: AskBio GmbH, Heidelberg, Germany. [10]Present address: Institute of Physics, École Polytechnique Fédérale de Lausanne (EPFL), Lausanne, Switzerland. ✉e-mail: valerie.doye@ijm.fr; martin.beck@biophys.mpg.de

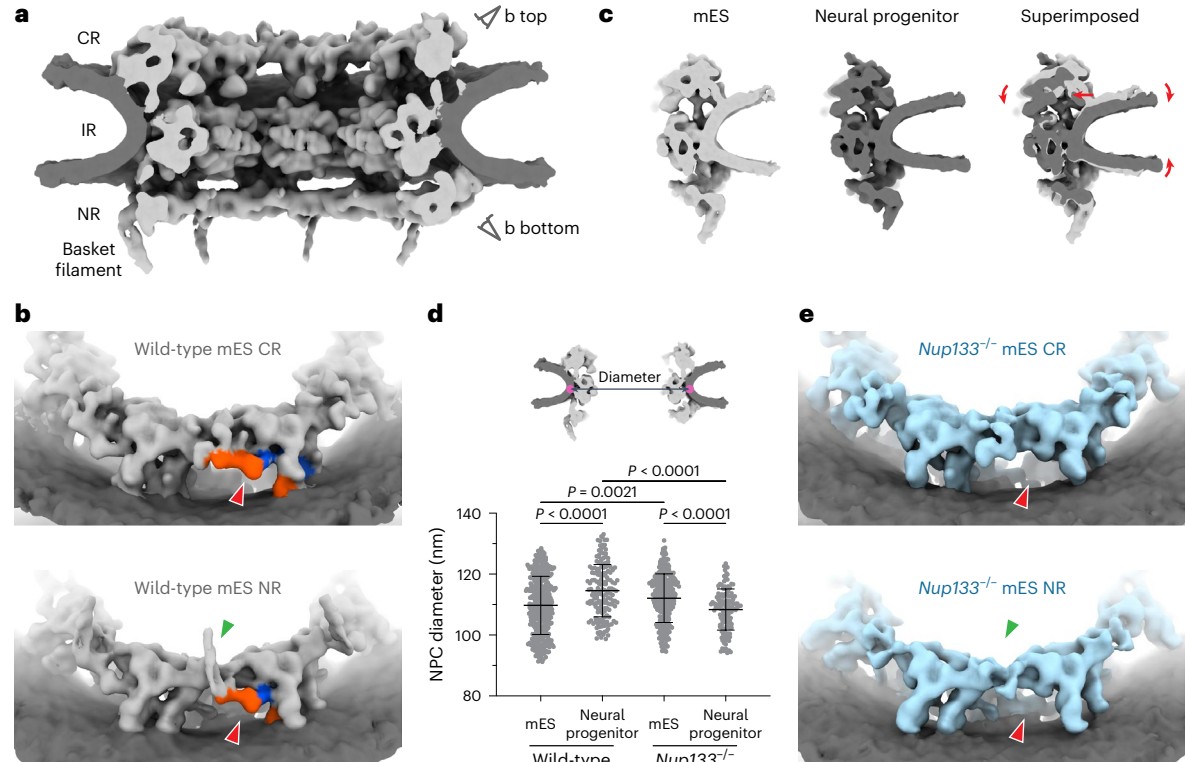

**Fig. 1 | Architectures of mES and neural progenitor NPCs. a**, Composite cryo-EM map of the wild-type mES NPC, shown as a cutaway view. Viewing angles in **b** are indicated by eye symbols. **b**, Cryo-EM maps of the CR (top) and NR (bottom) of the wild-type mES NPC. Positions of outer (orange) and inner (blue) Nup133 are shown. **c**, Structural comparison of wild-type mES cells and neural progenitor NPCs. A single subunit is shown as cutaway side view. Two maps are superimposed based on the position of the IR protomer, and the relative shift of the CR and nuclear membrane in the neural progenitor NPC map in comparison to the mES NPC map is indicated by red arrows. **d**, NPC diameter measurements based on subtomogram averages. A schematic (top) illustrates the measured distance between two opposing IR protomers. The graph (lower panel) depicts the measured NPC diameters, with means and s.d. (black bars) shown ($n$ = 446 NPCs (wild-type mES), $n$ = 173 NPCs (wild-type neural progenitor), $n$ = 317 NPCs ($Nup133^{-/-}$ mES), $n$ = 138 NPCs ($Nup133^{-/-}$ neural progenitor)). The Kruskal–Wallis rank sum test followed by Dunn's multiple comparisons test was applied. **e**, Cryo-EM maps of the CR (top) and NR (bottom) of the $Nup133^{-/-}$ mES NPC, shown as in **b**. In **b** and **e**, the positions of Nup133 and the basket filament are highlighted with red and green arrowheads, respectively.

compared with isolated nuclear envelopes[9], where mechanical forces are alleviated. Thus, nuclear envelope membrane tension has been proposed to regulate NPC diameter[24]. Intriguingly, NPC diameter changes are suggested to be relevant to nuclear mechanosensing. The nucleus is physically linked to the cytoskeleton network[28], and mechanical forces sensed by the cytoskeletal filaments are transmitted to the nucleus and impose mechanical stress[29–31]. This in turn triggers nuclear mechanosensing responses, such as nuclear import of the transcription factor YAP (yes-associated protein 1)[32]. Nuclear mechanosensing broadly alters nucleocytoplasmic transport capacity[33], which is thought to be mediated by increased NPC permeability due to their tension-induced deformation[32,33].

Nuclear mechanosensing is involved in multiple biological processes, such as cell differentiation and development. For example, cell differentiation depends on mechanosensing of matrix stiffness[34]. Cell differentiation also involves the remodelling of nuclear properties, including nuclear stiffening[35,36] and cytoskeleton-mediated nuclear shaping[37]. These previous findings highlight the importance of nuclear mechanosensing during cell differentiation, and indicate that the mechanical load on the nuclear envelope increases during differentiation. Thus, it is conceivable that NPC architecture is also affected, but this possibility has not yet been examined.

In this Article we hypothesize that the NPC scaffold conformationally responds to the changing mechanical environment of the nuclear envelope during differentiation, and that such a response may be impaired when the NPC scaffold architecture is perturbed. To address whether such a link between NPC diameter change, cell differentiation and perturbation of the NPC scaffold architecture exists, we used cryo-electron tomography (ET) and compared NPC architectures in wild-type and Nup133-deficient cell lines, and in pluripotent and differentiated neural progenitor states.

## Results

### NPCs dilate during neuronal differentiation of mES cells

To investigate possible changes in NPC architecture during cell differentiation, mES cells were differentiated into neural progenitor cells as previously described (Extended Data Fig. 1a–c)[38]. Consistent with previous reports, most of the neural progenitor cells expressed the marker Pax6 (Extended Data Fig. 1d–f) and had lost pluripotency marker Oct3/4 expression (Extended Data Fig. 1g,h), indicating largely successful differentiation. To improve the morphological consistency of vitrified samples, mES cells were synchronized in early S-phase, where the nuclear size and DNA content are more homogeneous. Clusters of cells were dissociated by trypsinization, pipetted onto electron microscopy (EM) grids, plunge-frozen and subjected to specimen-thinning by cryo-focused ion beam (cryo-FIB) milling (Extended Data Fig. 2a–d). We acquired cryo-electron tomograms of the wild-type mES and neural progenitor cells, and structurally analysed their NPCs by subtomogram averaging (Extended Data Fig. 2e,f and Supplementary Table 1; details are provided in the Methods).

The subtomogram averages of NPCs from wild-type mES cells were resolved to a resolution of ~30 Å (Fig. 1a,b and Extended Data Fig. 3). The overall architecture of the wild-type mES NPC is highly reminiscent of that of human NPCs, including the density of Nup133

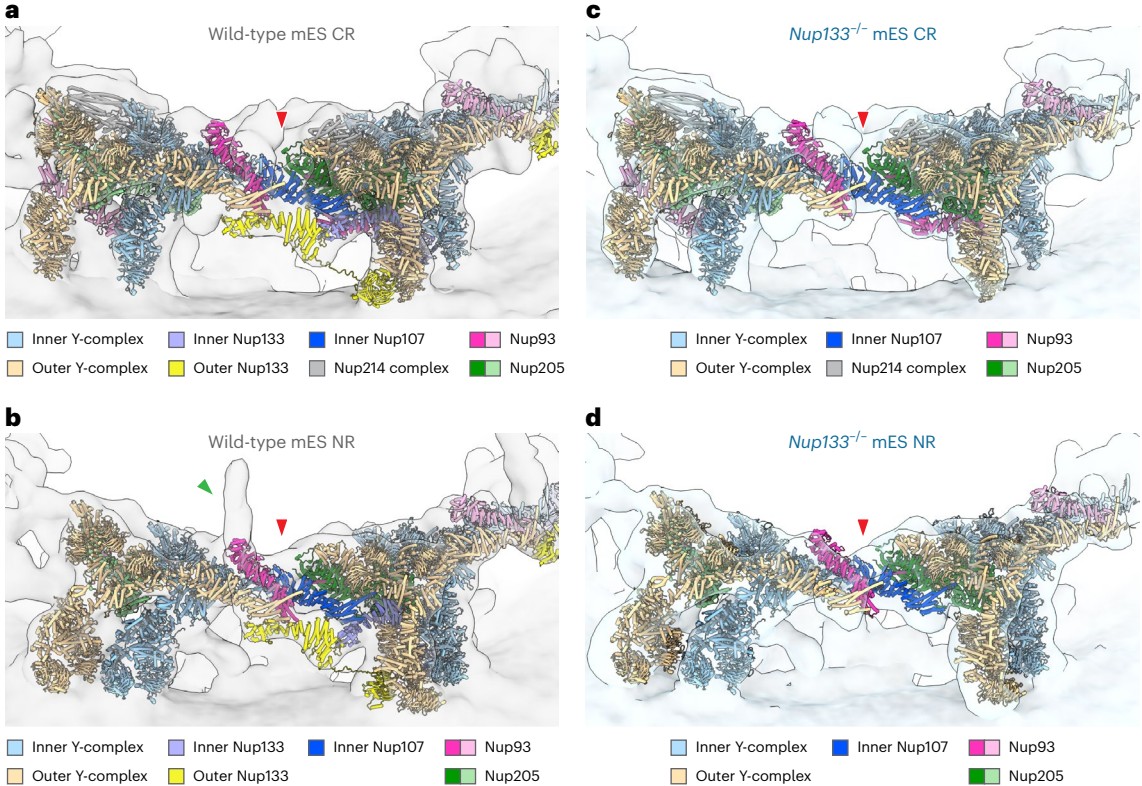

**Fig. 2 | The density for the head-to-tail contact is present in the *Nup133*$^{-/-}$ mES NPC. a–d**, The head-to-tail contact between the CR (**a**) and NR (**b**) protomers of the wild-type mES NPC, and the CR (**c**) and NR (**d**) protomers of the *Nup133*$^{-/-}$ mES NPC. The CR and NR models of dilated human NPC (PDB 7R5J) are fitted into the corresponding cryo-EM maps. Nup93 and Nup205 molecules involved in the

head-to-tail contact at the centre of the images are highlighted in bold colours. The main interaction interfaces are highlighted with red arrowheads. In **a** and **c**, the model of Nup358 is omitted for clarity. In **b**, the basket filament position is highlighted (green arrowhead).

(Extended Data Fig. 4). In addition to the structural features previously resolved in human NPC maps, a protrusion from the NR, probably corresponding to nuclear basket filaments, is resolved in the Nup107/Nup133 region (Fig. 1a,b and Extended Data Fig. 4b). Cryo-EM maps from differentiated neural progenitor cells also showed NPC structures almost identical to those from mES cells (Fig. 1c and Extended Data Fig. 3c), indicating that the NPC subunit composition remains largely similar during early neuronal differentiation, at least up to the neural progenitor state. However, compared to mES NPCs, the cryo-EM map of the neural progenitor NPCs shows an inward movement of the CR and higher nuclear envelope curvature at the nuclear membrane fusion point (Fig. 1c), both of which are indicative of NPC dilation[24,27]. Consistent with the inward movement of the CR, the angle formed between the membrane anchor sites of CR, IR and NR is wider in the neural progenitor than mES NPCs (Extended Data Fig. 3f,g), further supporting our structural observation. Measurements of individual NPC diameters based on subtomogram averages of opposing subunits (see Methods for details) indeed confirm that wild-type neural progenitor NPCs have a substantially larger diameter than wild-type mES NPCs (Fig. 1d), supporting our notion that NPC architecture is affected during cell differentiation. Notably, the nuclear envelope in neural progenitor cells showed reduced thickness in comparison to mES cells (Extended Data Fig. 2g–i), which would, for example, occur when the nuclear envelope is laterally stretched without luminal volume change. As NPCs are embedded in the nuclear envelope, any lateral membrane stretching would be propagated to them and could induce conformational changes of the complex. Thus, the reduced nuclear envelope thickness observed in the neural progenitor cells is also in line with the dilation of the wild-type neural progenitor NPCs.

### *Nup133*$^{-/-}$ NPCs retain remnant contacts between protomers
Nup133 mediates the head-to-tail connection between adjacent Y-complexes and bridges neighbouring protomers of the CR and NR[9,11]. Despite this critical scaffolding role[5], Nup133 is dispensable for mES cell proliferation[19,20,22]. However, its absence is detrimental for terminal differentiation to postmitotic neurons, and results in reduced growth and increased cell death upon differentiation induction[19,22]. We therefore reasoned that *Nup133*$^{-/-}$ mES cells would be a suitable model system to further assess the importance of the Y-complex for the structural integrity of the NPC architecture during differentiation.

We structurally analysed NPCs in previously generated *Nup133*$^{-/-}$ mES cells[20] by subtomogram averaging. As expected, cryo-EM maps of the *Nup133*$^{-/-}$ mES NPC lack the density for Nup133 in the CR and NR (compare Fig. 1b,e), whereas the IR shows a structure almost identical to that of the wild-type mES NPC (Extended Data Fig. 3b–d). Strikingly, even without Nup133, the CR and NR appear to have an overall intact structure. A connection between the adjacent protomers is observed in the Nup107–Nup205/Nup93 heterodimer region in a manner consistent with that in the wild-type mES NPC (Fig. 2), highlighting a previously underappreciated importance of this additional head-to-tail contact interface. This interaction could be particularly important for the NR, as additional components that would support structural integrity, such as cytoplasm-specific Nup358, are absent. In accordance with the previously reported nuclear basket misassembly and increased dynamics of a basket component in *Nup133*$^{-/-}$ cells[20], the respective protrusion in the Nup107/Nup133 region was diminished (compare Fig. 1b,e; Fig. 2).

### *Nup133*$^{-/-}$ NPCs constrict during neuronal differentiation
Nup133-deficient cells have been reported to differentiate into neural progenitor cells, albeit with abnormally maintained characteristics of

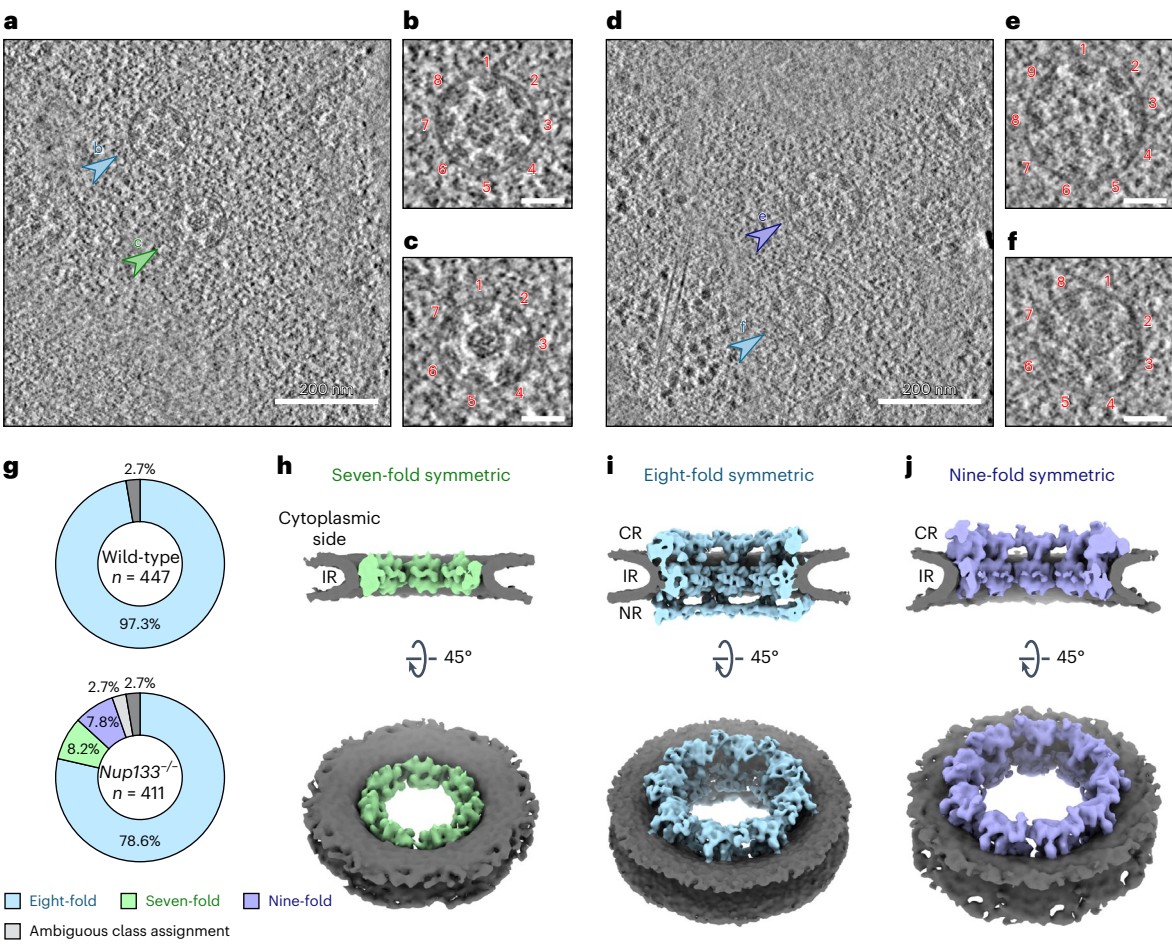

**Fig. 3 | NPCs with non-canonical symmetries are present in *Nup133*⁻/⁻ mES cells.** a–f, Representative slices from reconstructed tomograms of the *Nup133*⁻/⁻ mES cells, showing top views of seven-fold (**a,c**), eight-fold (**a,b,d,f**) and nine-fold (**d,e**) symmetric NPCs. In **a,d**, the eight-fold symmetric NPCs are indicated by light blue arrowheads, and NPCs with seven-fold (**c**) and nine-fold (**e**) symmetric architectures are indicated by green and purple arrowheads, respectively. The top views in **a** and **d** are shown as enlarged views in **b**, **c**, **e** and **f**, with protomers numbered. Particle numbers are provided in Supplementary Table 1. Scale bars, 50 nm (**b,c,e,f**). **g**, Pie chart of the particle distribution after reference-based classification. Particles with ambiguous class assignment are indicated in light grey, and particles discarded before classification are in dark grey. **h–j**, Composite cryo-EM maps of the seven-fold (**h**), eight-fold (**i**) and nine-fold (**j**) symmetric NPCs, shown as a cutaway view (top) and a cytoplasmic view (bottom). Note that the CR and NR of the seven-fold symmetric NPC, as well as the NR of the nine-fold symmetric NPC did not yield interpretable subtomogram averages and are not included in the composite cryo-EM maps in **h** and **j**.

the pluripotent state[19], and ultimately fail in terminal differentiation to postmitotic neurons[19,22]. We therefore structurally analysed the NPC architecture in neural progenitor cells obtained from *Nup133*⁻/⁻ mES cells (Extended Data Fig. 1d,f,h). Subtomogram averages of the NPC from *Nup133*⁻/⁻ neural progenitor cells were reminiscent of those of the *Nup133*⁻/⁻ mES NPC (Extended Data Fig. 3d,e). Surprisingly, however, the diameter measurements revealed a constriction, rather than a dilation, of the *Nup133*⁻/⁻ neural progenitor NPCs in comparison to the *Nup133*⁻/⁻ mES NPCs (Fig. 1d). This is in contrast to the wild-type datasets, where we observed NPC dilation within the differentiated neural progenitor cells. Notably, similar to the wild-type dataset, the thickness of the nuclear envelope in the *Nup133*⁻/⁻ neural progenitor cells was significantly reduced in comparison to mES cells (Extended Data Fig. 2i), indicating that the potential lateral stretching of the membranes also occurs in this background. Because NPCs cannot actively change their diameter and rather passively react to external mechanical forces[24], these data imply that the effects of differentiation on nuclear envelope properties, such as nuclear envelope stretching, are improperly propagated to the NPCs in *Nup133*⁻/⁻ neural progenitor cells. Although we cannot yet directly measure nuclear membrane tension in our samples, these findings point towards a potential link

between Y-complex structural integrity and the mechanical properties of the nuclear envelope itself.

## A subset of *Nup133*⁻/⁻ NPCs exhibit non-canonical symmetries

To understand how Nup133 loss is linked to nuclear envelope physical properties, we further characterized NPCs in *Nup133*⁻/⁻ mES cells. As previous studies reported basket assembly defects among subsets of *Nup133*⁻/⁻ mES NPCs[19,22], we aimed to determine whether potential subclasses of NPCs with distinct structural features exist. Visual inspection of the reconstructed tomograms revealed NPCs with non-canonical seven- or nine-fold symmetric architectures in the tomograms from *Nup133*⁻/⁻ mES cells, in addition to the canonical eight-fold symmetric NPCs (Fig. 3a–f). This is surprising, because non-eight-fold symmetric NPCs are thought to be very rare[39]. By reference-based classification (see Methods for details), 34 and 32 out of 400 particles were classified as seven-fold and nine-fold symmetric NPCs, respectively (Fig. 3g and Extended Data Fig. 5a). Similar classification using the particles of the wild-type mES NPC resulted in one class with eight-fold symmetric architecture (Fig. 3g and Extended Data Fig. 5b), indicating that NPCs with aberrant symmetries are rarely observed in wild-type mES cells.

We further characterized NPCs with non-canonical symmetries using subtomogram averaging. The particle sets of seven-fold and nine-fold symmetric NPCs yielded moderately resolved averages of the IRs (Extended Data Fig. 5c,d). Although the resolution is reduced due to the limited number of particles, the observed structural features are consistent overall with canonical IRs (Fig. 3h–j). In contrast, the averages of the CRs and NRs of seven-fold symmetric NPCs did not exhibit any interpretable structural features (Extended Data Fig. 5c). Averages of nine-fold symmetric NPCs also failed to show clear NR-like architecture (Extended Data Fig. 5d), although structural features typical of the CR were apparent (Fig. 3j and Extended Data Fig. 5d). These observations indicate that the subpopulation of NPCs with aberrant symmetries possess intact IRs, whereas CRs and NRs are in part deteriorated or diminished. The observed remnant densities for CRs and NRs (Extended Data Fig. 5c,d) may be indicative of further structural heterogeneity. However, given the low particle number, these cannot be further analysed by averaging-based methods such as subtomogram averaging.

### Nup133[−/−] NPCs are heterogeneous, with incomplete rings

Three-dimensional (3D) template matching (TM)[40,41] can be used to analyse structural heterogeneity. TM detects the structural signature of target molecules within cryo-electron tomograms by cross-correlating a reference structure with all possible locations and orientations of a given tomogram, without the need for averaging. We recently showed that this method can detect individual NPC subunits[42], thus opening up the possibility of examining the heterogeneous presence of NPC subunits. We used cryo-EM maps of the CR, IR and NR protomers as search templates and analysed the ultrastructure of individual NPCs with non-canonical symmetry in the Nup133[−/−] mES dataset (Fig. 4a). In these selected particles, IR protomers were detected as complete or almost complete rings with the expected seven- or nine-fold symmetries (Extended Data Fig. 6a,b). However, neither CR nor NR protomers were detected in almost half of the seven-fold symmetric NPCs, indicating a complete absence of the respective rings (Fig. 4b,c). The remaining particles showed the presence of two to five CR protomers (Fig. 4c) arranged in a partially open ring-like architecture, whereas NR protomers were less frequently detected (Fig. 4c). In nine-fold symmetric NPCs, one-third of the tested particles showed complete CRs, and the rest contained five to eight protomers arranged in an incomplete ring architecture (Fig. 4d,e). The number of detected NR protomers was more variable, yet the majority of the nine-fold symmetric NPCs had a partially open NR architecture (Fig. 4e).

We next analysed NPC subsets of similar size with canonical eight-fold symmetry. In the Nup133[−/−] cells, only 4 out of 24 NPCs showed a complete CR and NR, and the majority of the particles still had incomplete architectures (Fig. 5a,b). This is in contrast to wild-type mES NPCs, in which 18 out of 29 particles have fully detectable CRs and NRs, and most of the remaining 11 have close to complete ring architectures (Fig. 5c,d). In both datasets, the IR protomers were similarly detected as complete or almost complete eight-fold symmetric rings (Extended Data Fig. 6c,d). These results clearly show that eight-fold symmetric NPCs devoid of Nup133 more frequently exhibit heterogeneous and incomplete CRs and NRs than wild-type NPCs, and that the incomplete ring architectures can be observed globally among Nup133[−/−] NPCs, independent of their symmetries. We thus conclude that the lack of Nup133 globally affects the structural integrity and completeness of CRs and NRs in addition to perturbing the symmetry of a smaller subpopulation of NPCs. Importantly, the TM analysis revealed heterogeneity in the NPC architecture on the single protomer level, which would have been overlooked with averaging-based particle analysis.

### NPC over-stretching results in nuclear envelope openings

Nuclear envelope tension has been proposed to regulate NPC dilation, and in its absence, NPCs constrict into a conformational ground state[24]. In such a scenario, one may conceptualize the NPC scaffold as an annular spring that counteracts laterally applied forces and thereby maintains the pore size within a certain range. During NPC dilation, CRs and NRs are probably conformationally strained (Fig. 1c)[10,27], and may thus be regarded as the source of energy to bring the NPC architecture back into a conformational ground state. We therefore reasoned that NPCs with an incomplete ring architecture may have at least partially lost the spring-like properties, and further hypothesized that they may excessively dilate (over-stretch) and possibly disintegrate during differentiation due to having insufficient structural support to withstand the mechanical stress. If a certain subpopulation of NPCs in the nuclear envelope were over-stretched, nuclear envelope membrane tension could be relieved overall, and this globally reduced tension could in turn allow the remaining NPCs to constrict (Fig. 6a). This model could explain the NPC constriction we observed in Nup133[−/−] neural progenitors in subtomogram average-based diameter measurements (Fig. 1d). Moreover, the model leads to several predictions: (1) large openings in the nuclear envelope should represent over-stretched NPCs and thus may contain NPC scaffolds or parts thereof; (2) over-stretched NPCs should occur more frequently upon differentiation than in the pluripotent state; and (3) over-stretched NPCs should occur more frequently in Nup133[−/−] cells than in wild-type cells.

To test the first prediction, that incomplete NPCs over-stretch and create large openings in the nuclear envelope, we examined the tomograms of neural progenitor cells obtained from Nup133[−/−] cells, where the aberrant NPCs with an incomplete architecture would probably have been exposed to the increased nuclear envelope mechanical stress. Indeed, we frequently found abnormally large holes in the nuclear envelope (Fig. 6b and Extended Data Fig. 7a,b), which are inconsistent with structural models of intact NPCs and thus were not included in the subtomogram averaging-based NPC diameter measurement (Fig. 1d). In 19 of the 24 analysed large membrane holes, TM analysis detected subunits of at least one of the three rings (Extended Data Fig. 7a), confirming that the observed membrane holes are indeed lined with remnants of NPCs, presumably over-stretched and already largely disintegrated. At these sites, incomplete CRs are often detected at a position distant from the nuclear envelope (Fig. 6b and Extended Data Fig. 7a), implying that they have detached from the membrane. Such membrane dissociation is less prominently observed for NRs (Extended Data Fig. 7a), possibly reflecting a stronger membrane association supported by the NR-specific component ELYS[10]. In addition,

**Fig. 4 | TM analysis of the seven-fold and nine-fold symmetric NPCs shows incomplete CR and NR ring architectures. a**, Schematic showing the overall TM analysis workflow. Subvolumes containing NPCs of interest are extracted from tomograms and subjected to three TM runs, using the CR, IR and NR averages as the search templates. Ellipsoidal masks used for the TM runs are shown as grey transparent spheres around the averages. The positions of the peaks were extracted from the output cross-correlation (CC) score volumes (see Methods for details). For visualization of the peaks, ±15 z-stacks around the cross-correlation peaks are extracted, and maximum values along the z axis are presented as 3D plots. Based on the positions of the extracted peaks and corresponding template orientations, the maps of the templates are projected back to generate a pseudo-composite map (rightmost panel). **b**, Representative results of the TM analysis of a seven-fold symmetric NPC. 3D plots of the peaks and a pseudo-composite map are shown as in **a**. **c**, 3D histogram showing the number of seven-fold symmetric NPCs and their differing numbers of CR and NR peaks detected by the TM analysis (n = 24 NPCs). **d**, Representative results of the TM analysis of a nine-fold symmetric NPC. 3D plots of the peaks and pseudo-composite map as in **a**. **e**, Analysis as in **c** for the nine-fold symmetric NPCs with different numbers of CR and NR peaks (n = 20 NPCs). In the pseudo-composite maps in **a** and **d**, gaps in the ring architectures are indicated by red arrowheads. In **c** and **e**, bars that include the examples shown in **a**, **b** and **d** are indicated by green arrowheads.

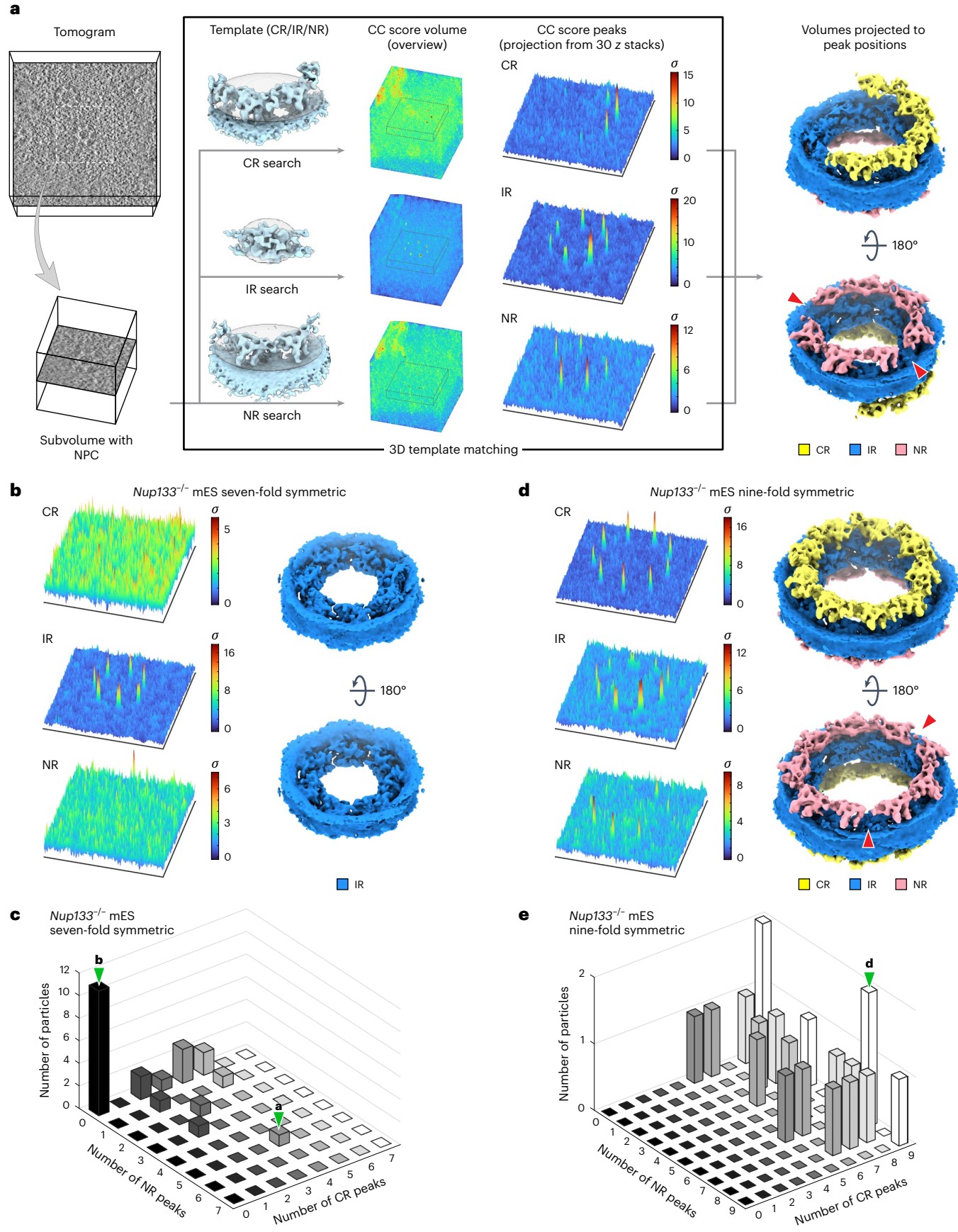

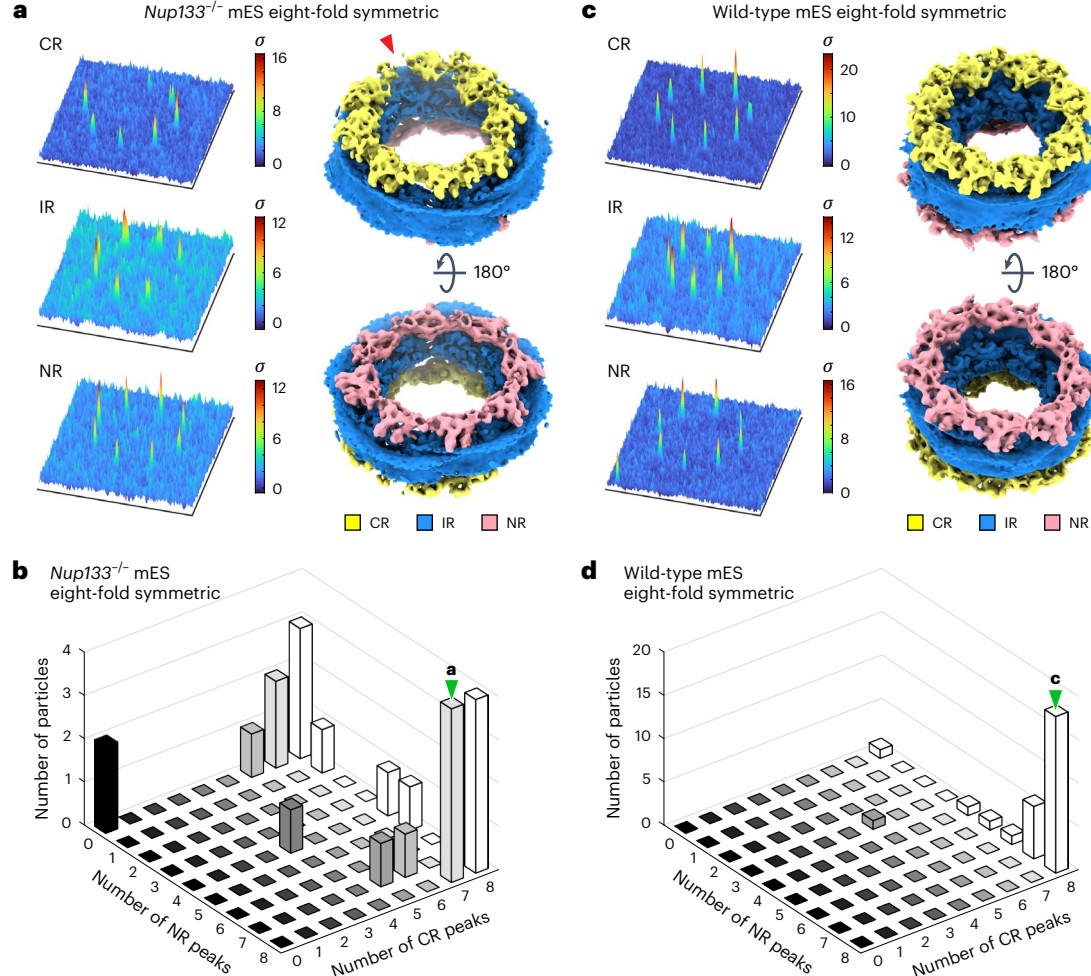

**Fig. 5 | Eight-fold symmetric NPCs in the *Nup133⁻/⁻* mES cells more frequently possess incomplete CR and NR architectures than wild-type mES NPCs.** **a**, Representative results of TM analysis of an eight-fold symmetric NPC in *Nup133⁻/⁻* mES cells. **b**, 3D histogram showing the number of eight-fold symmetric *Nup133⁻/⁻* mES NPCs and their differing numbers of CR and NR peaks (*n* = 24 NPCs). **c**, Representative results of the TM analysis of an eight-fold symmetric NPC in wild-type mES cells. **d**, Analysis as in **b** for the eight-fold symmetric wild-type mES NPCs (*n* = 29 NPCs). In **a** and **c**, 3D plots of the peaks and pseudo-composite map are as in Fig. 4a. In the pseudo-composite map in **a**, a gap in the ring architecture is indicated with a red arrowhead. In **b** and **d**, bars that include the examples shown in **a** and **c** are indicated by green arrowheads.

the IRs in these aberrant NPCs are always incomplete, and individual IR protomers are often distantly spaced (Extended Data Fig. 7a), which is inconsistent with an intact linker connection between two adjacent IR protomers. Taken together, these structural features indicate that the observed membrane holes are NPCs that have disintegrated, probably due to over-stretching.

### *Nup133⁻/⁻* NPCs frequently disintegrate upon differentiation
We next examined our model prediction (2) and asked whether the over-stretched NPCs were more frequent in the neural progenitor cells. As the initial subtomogram average-based NPC diameter measurement (Fig. 1d) only included NPCs with canonical particle sizes and apparently intact architecture, we revisited the reconstructed tomograms and measured the size of all the membrane openings in the nuclear envelopes. The initially measured NPC diameters based on subtomogram averages were distributed between 90 nm and 135 nm (Fig. 1d). We thus set an arbitrary threshold for the diameter of unperturbed NPCs to a maximum of 135 nm, whereby membrane openings larger than 135 nm were considered over-stretched or disintegrated NPCs. As expected, the frequency of membrane openings larger than this cutoff was elevated in the wild-type neural progenitor over wild-type mES cells (Fig. 6c). This is consistent with the observation that NPCs

are dilated in neural progenitor cells (Fig. 1d) and the hypothesis that this dilation is probably driven by increased membrane tension[24], which is also supported by the reduced nuclear envelope thickness in neural progenitor cells (Extended Data Fig. 2g–i). To address our last prediction, that over-stretched NPCs occur more frequently in Nup133-deficient cells, we examined the *Nup133⁻/⁻* mES cells, and found that the frequency of large membrane openings is indeed strongly increased compared to wild-type mES cells and reaches 20% (Fig. 6c). Although this frequency remains similar in *Nup133⁻/⁻* neural progenitor cells, the distribution of the diameter shows that the size of the membrane openings generally becomes larger and more variable (Fig. 6d), indicating that the NPC architecture is more adversely affected upon differentiation. In line with this, membrane openings larger than 150 nm, which we consider largely disintegrated NPCs, more frequently exist in *Nup133⁻/⁻* neural progenitor cells than *Nup133⁻/⁻* mES cells (16% in the neural progenitor dataset, 9.2% in the mES dataset). Together, we speculate that an increase in nuclear envelope tension arising from changes in nuclear morphology during differentiation can lead to NPC over-stretching and disintegration in *Nup133⁻/⁻* cells. Although the exact causal relationships, including formal proof for an increase in nuclear membrane tension, await further investigation, the frequent NPC disintegration observed in *Nup133⁻/⁻* cells could well

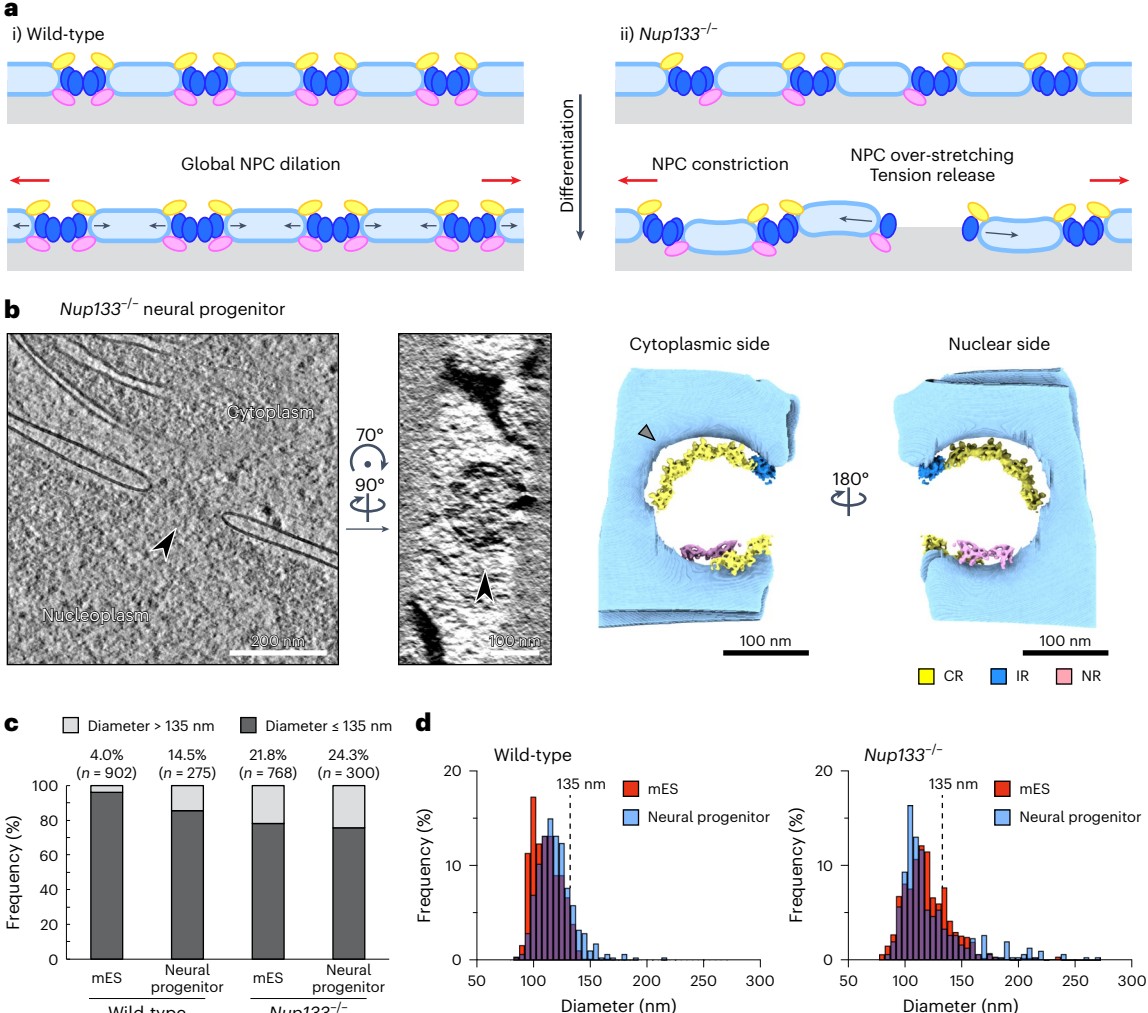

**Fig. 6 | Over-stretched NPCs are present in *Nup133*⁻/⁻ neural progenitor cells.** **a**, Hypothetical model of NPC diameter changes during differentiation: (i) in wild-type cells, the increased nuclear envelope tension during early neuronal differentiation is homogeneously propagated along the nuclear envelope, leading to overall dilation of the NPCs; (ii) in *Nup133*⁻/⁻ cells, the increased nuclear envelope tension causes over-stretching of NPCs with impaired structural robustness, leading to release of the nuclear envelope tension and overall constriction of the NPCs. The nuclear envelope is coloured in light blue, and the CR, IR and NR of the NPC are coloured in yellow, blue and pink, respectively. Red and black arrows depict nuclear envelope membrane tension and the motion of NPC scaffolds. **b**, Representative example of the over-stretched NPCs observed in the *Nup133*⁻/⁻ neural progenitor dataset: slices from the reconstructed tomograms showing the side views (left) and top views (middle) of the over-stretched NPCs, together with pseudo-composite maps generated from the results of the TM analysis (right). CRs, IRs and NRs in the pseudo-composite maps are coloured as in Fig. 3a and are shown with the segmented membrane (light blue). The site of CR detachment from the outer nuclear membrane is highlighted with a grey arrowhead. All analysed over-stretched NPCs are shown in Extended Data Fig. 7. **c**, Quantification of the over-stretched NPCs in the four datasets (wild-type mES and neural progenitor cells; *Nup133*⁻/⁻ mES and neural progenitor cells). The number of analysed NPCs and the percentage of over-stretched NPCs (diameter > 135 nm) are indicated on top of each bar. **d**, Histogram showing the distribution of the measured NPC diameters depicted in **c**. Note that the overall diameter distribution is consistent with the results of subtomogram average-based measurements in Fig. 1d.

be a contributing factor to the previously described differentiation defect phenotype[19,22].

### *Nup133*⁻/⁻ neuronal precursors show nuclear integrity defects

In our experimental set-up, neural progenitor cells were obtained by culturing mES cells on a non-adhesive dish. To obtain mature neurons, these neural progenitor cells are seeded on an adhesive surface and further cultured under neuronal differentiation-inducing conditions. We thus reasoned that the increased frequency of over-stretched and disintegrated NPCs in the *Nup133*⁻/⁻ neural progenitor cells may impair this subsequent neuronal differentiation process, where cell adhesion and nuclear shaping would be involved. In particular, NPC over-stretching and disintegration could alter nucleocytoplasmic compartmentalization and affect permeability barrier integrity.

Nuclear DNA damage has been reported to increase when nuclear integrity is compromised[43–46]. We thus assessed steady-state DNA damage levels in neural progenitor cells undergoing neuronal differentiation (hereafter referred to as neuronal precursors) by quantifying phosphorylated histone H2AX (γ-H2AX) and 53BP1 foci, indicators of DNA damage and repair[47–50]. We observed an increase in foci in *Nup133*⁻/⁻ neuronal precursors compared to wild-type cells 24 h after induction of neuronal differentiation (Fig. 7a–c). No statistically significant difference between the two genetic backgrounds was present in the mES cells (Extended Data Fig. 8a–c), nor in neuronal precursors 2 h after neuronal differentiation induction (Extended Data Fig. 8d–f), indicating that the effect is probably linked to changes occurring during neuronal differentiation. Moreover, wild-type and *Nup133*⁻/⁻ cells showed comparable sensitivity to DNA-damaging agents, both in the pluripotent

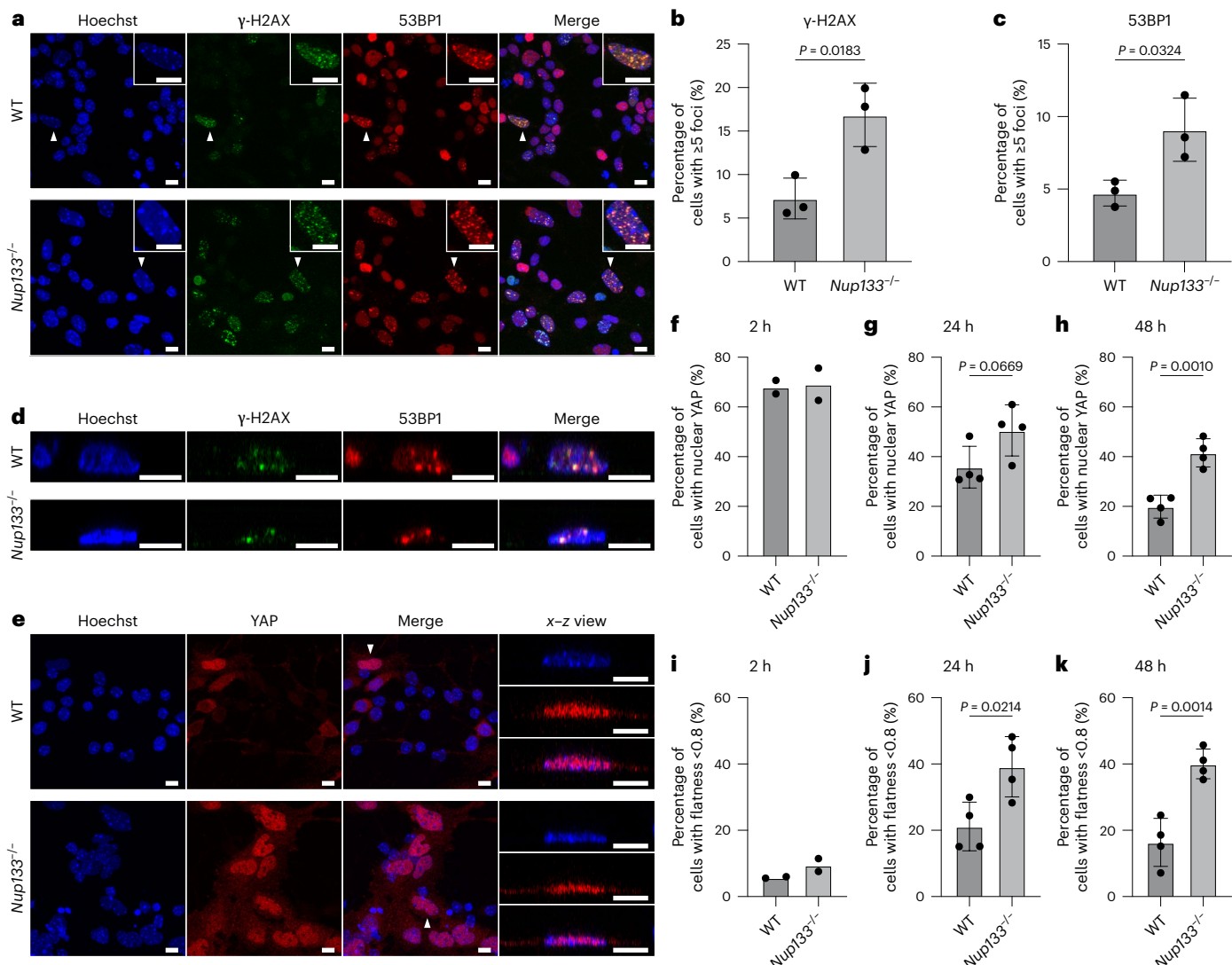

**Fig. 7 | Accumulation of DNA damage is observed in *Nup133*⁻/⁻ neuronal precursors. a**, Representative immunofluorescence staining images of neuronal precursors with γ-H2AX and 53BP1 foci 24 h after neuronal differentiation induction. Maximum projection images are shown. Examples of nuclei with ≥5 detected foci are indicated with white arrowheads and highlighted in the insets. Scale bars, 10 μm. **b,c**, Mean percentage of nuclei with ≥5 γ-H2AX (**b**) and 53BP1 (**c**) foci in wild-type and *Nup133*⁻/⁻ neuronal precursors. Data are from three biological replicates, and at least 330 nuclei were analysed for each replicate. Each data point represents the percentage calculated from the individual dataset. Error bars denote s.d. Two-tailed Student's *t*-test. **d**, Representative *x–z* orthogonal views of nuclei from neuronal precursors with γ-H2AX and 53BP1 foci. The nuclei highlighted with white arrowheads in **a** are shown. Scale bars, 10 μm. **e**, Representative immunofluorescence staining images of neuronal precursors

with nuclear YAP 48 h after neuronal differentiation induction. Maximum projection images are shown together with *x–z* orthogonal views of the nuclei highlighted with white arrowheads. Scale bars, 10 μm. **f–h**, Mean percentage of nuclei with nuclear YAP localization in neuronal precursors at 2 h (**f**), 24 h (**g**) and 48 h (**h**) time points after neuronal differentiation induction. Data are from two (**f**) or four (**g,h**) biological replicates, and at least 340 nuclei were analysed for each replicate. In **g** and **h**, error bars denote s.d. Two-tailed Student's *t*-test. **i–k**, Mean percentage of nuclei with flatness index below 0.8 in neuronal precursors at 2 h (**i**), 24 h (**j**) and 48 h (**k**) after neuronal differentiation induction. Data are from two (**i**) or four (**j,k**) biological replicates, and at least 340 nuclei were analysed for each replicate. In **j** and **k**, error bars denote s.d. Two-tailed Student's *t*-test.

state and during the neuronal differentiation process (Extended Data Fig. 8g,h), strongly arguing against the possibility that increased foci in the *Nup133*⁻/⁻ neuronal precursors compared to the wild-type cells are due to major defects in the DNA damage response or DNA repair.

When analysing the neuronal precursors in detail, we noticed a morphological heterogeneity, where a subpopulation of cells contained flatter, more widespread nuclei (Fig. 7d). This phenotype appeared to be particularly frequent in cells with an increased number of γ-H2AX and 53BP1 foci. We therefore quantified the flatness index of nuclei 2 h and 24 h after neuronal differentiation induction, and found that it was indeed overall lower for cells with more than five

foci (Extended Data Fig. 9a–d). This was the case for both the wild-type and *Nup133*⁻/⁻ background and at both time points, although the phenotype was less pronounced at the 2 h time point (Extended Data Fig. 9a,b). This indicates that a flatter nuclear morphology coincides with an increase in DNA damage in the neuronal precursors and that the phenotype becomes more prominent over the course of neuronal differentiation. Moreover, at 24 h, the overall distribution of the flatness index was in general lower in the *Nup133*⁻/⁻ neuronal precursors than in the wild-type cells (Extended Data Fig. 9c,d), indicating that *Nup133*⁻/⁻ neuronal precursors are more prone to develop a flatter nuclear morphology during neuronal differentiation.

Physiologically, a flatter morphology could indicate an increase in mechanical strain on cells and nuclei. The mechanosensitive transcription factor YAP has been reported to relocalize to the nucleus when nuclei-flattening forces are applied[32]. Increased nuclear import of YAP has been proposed to result from NPC deformation and increased permeability[32,33], as the nuclear membrane is stretched and curved. Thus, to further characterize the nuclear morphology change we observed, we also analysed nuclear YAP intensity in these cells. As expected, cells with flat nuclear morphology exhibited an overall higher nuclear YAP intensity at all the analysed time points (Fig. 7e and Extended Data Fig. 9e–i), indicating that the nuclei with flat morphology might indeed be mechanically more strained. In addition, the protein levels of YAP as well as its nuclear import are known to be negatively regulated during neuronal differentiation[51,52]. Consistently, a high proportion of wild-type and *Nup133*[−/−] cells exhibited nuclear YAP staining at 2 h after neuronal differentiation induction, and the frequency was reduced at later time points in the wild-type neuronal precursors (Fig. 7e–h and Extended Data Fig. 9h,i). In contrast, the frequency of such YAP-positive nuclei remained high in the *Nup133*[−/−] neuronal precursors at both 24 h and 48 h when compared to wild-type cells (Fig. 7e–h and Extended Data Fig. 9i), and more nuclei showed a flatter morphology (Fig. 7i–k). Overall, these observations indicate that the flatter nuclear morphology and sustained level of nuclear YAP in *Nup133*[−/−] compared to wild-type neuronal precursors are phenotypes that arose during neuronal differentiation. Although a dissection of the direct causal relationships remains to be done, our findings point to the possibility that Nup133 loss-induced abnormalities in nuclear envelope integrity and mechanical properties may become functionally relevant in conditions where mechanical strain probably occurs, for example, during neuronal differentiation.

### NPC over-stretching is not specific to mouse *Nup133*[−/−] cells

Within the tomograms of the wild-type neural progenitor, we found a subset of NPCs that exhibit abnormally large diameters, indicating that over-stretching of NPCs could occur even without perturbation of the NPC scaffold architecture (Fig. 6c,d). To better understand this process, we further analysed the architectures of the over-stretched NPCs in wild-type neural progenitor cells using TM. Similar to those in *Nup133*[−/−] neural progenitor cells (Fig. 6b), over-stretched NPCs in wild-type neural progenitor cells often have their CRs at a position distant from the nuclear membranes (Fig. 8a and Extended Data Fig. 10a). However, in contrast to over-stretched NPCs in *Nup133*[−/−] neural progenitor cells, where CR architectures are largely impaired (Extended Data Fig. 7a), we frequently observed a complete or almost complete eight-fold symmetric CR (Fig. 8a and Extended Data Fig. 10a). Complete rings are observed mostly for the CR, whereas the NR of over-stretched NPCs is more often fragmented (Extended Data Fig. 10a), consistent with the idea that the CR has a more resilient architecture owing to the additional cytoplasm-specific components such as Nup358. Notably, the IRs in these over-stretched NPCs are incomplete and arranged in a distorted manner, indicating that their architectures are largely impaired.

Finally, we asked whether over-stretched NPCs can be found in other cell types and species, and turned to a cryo-ET dataset of primary human macrophages[53]. These cells are mobile and strongly adherent to the respective surfaces, implying that their nuclei are exposed to substantial mechanical stress. In cryo-ET images from adherent macrophages, we indeed found a small number of NPCs with diameters of 140–150 nm (Fig. 8b and Extended Data Fig. 10b). TM analysis revealed that these potentially over-stretched NPCs possess structural features consistent with those observed in over-stretched murine NPCs in wild-type neural progenitor cells, namely a membrane-detached complete CR and partially impaired NR, although some of the NRs are also detached from the membrane (Fig. 8b and Extended Data Fig. 10b). Moreover, similar to the over-stretched murine NPCs, no complete IR was detected in the over-stretched NPCs from human macrophages (Fig. 8b and Extended Data Fig. 10b), possibly implying that the IR architecture is in general more sensitive to over-stretching than CR and NR. Taken together, we conclude that over-stretching of the NPC architecture occurs in a similar manner in different cell types with distinct cellular states.

## Discussion

Our study reveals the importance of a complete NPC scaffold for protecting the overall NPC architecture and integrity during cell differentiation, when increased mechanical force is imposed on the nucleus. Moreover, our findings provide a plausible model for NPC disintegration by excess dilation, for example, upon increased mechanical stress (Fig. 8c). In this model, the rigid ring architectures of the CR and NR of the NPC normally limit over-stretching and disintegration to a low frequency, which may prevent large-scale damage to the nuclear envelope (Fig. 8c,i). Only when the NPC is exposed to substantial stretching stress and is excessively over-stretched does it start to disintegrate, mainly by detachment of the CR from the nuclear membrane. In contrast, the NPCs without Nup133 lack the stable connection between the CR and NR protomers, resulting in the formation of incomplete CRs and NRs. These incomplete rings fail to sufficiently restrict NPC dilation, leading to increased over-stretching and disintegration of NPCs (Fig. 8c,ii).

We propose that the NPC can be considered a mechanical buffer that provides additional nuclear surface area under conditions of mechanical stress. In mammalian tissue culture cells, the NPC density is ~5–10 NPCs $\mu m^{-2}$ (refs. [54–56]), and NPCs constitute ~5–10% of the nuclear surface area. As a consequence of dilation from 90 nm to 130 nm (Fig. 1d), the area taken up by NPCs would roughly double and the nuclear surface area would increase by ~10%. Such an expansion capacity of NPCs would allow modulation of the nuclear surface area in response to external mechanical stimuli with reduced stress imposed on the lipid bilayer, and thereby act as a buffer for nuclear envelope stress without impairing nucleocytoplasmic compartmentalization. If the scaffold architecture is perturbed, the structural rigidity and plasticity of the NPC are lost, leading to loss of this buffering function and higher sensitivity of nuclei to mechanical stress.

The presence of a complete nine-fold symmetric CR architecture (Fig. 4d,e) in Nup133-deficient cells implies that the connection between two adjacent CR protomers tolerates wider angles when Nup133 is absent, further suggesting an increased flexibility of their head-to-tail contact. It is thus likely that Nup133 sterically limits the angles of the Y-complex head-to-tail contact and provides mechanical rigidity to these interfaces. Intriguingly, this steric restriction may also play an important role for the proper assembly of the NPC. Across species, the eight-fold symmetric architecture of the NPC is strictly conserved[57], and an NPC architecture with non-canonical symmetry has rarely been reported, with, to the best of our knowledge, only one EM study showing the presence of nine-fold and ten-fold symmetric NPCs within *Xenopus* oocytes at low frequencies[39]. Thus, the presence of seven-fold and nine-fold symmetric NPCs at a percentage of ~5% each within the *Nup133*[−/−] mES cells is a striking phenotype, and clearly indicates a perturbed NPC assembly due to the absence of Nup133. Because the NR is thought to serve as a primer for NPC biogenesis during both the postmitotic and interphase NPC assembly pathways[58], we speculate that the architecture of the Y-complex, especially its tail structured by Nup133, determines the arrangement, stoichiometry and spacing of the assembling NRs, which subsequently regulate the geometry of the IR and ultimately of the entire NPC. The absence of fully formed NR ring architectures in the seven-fold and nine-fold symmetric NPCs, as observed in our TM analysis (Fig. 4a–e), may suggest that a complete NR ring architecture with non-canonical symmetry is initially formed at the assembling NPC, but that it disintegrates afterwards, potentially due to unstable contacts between adjacent protomers.

In addition, our TM analysis revealed the presence of incomplete CRs and NRs, particularly within the *Nup133*[−/−] cells, and thereby raised

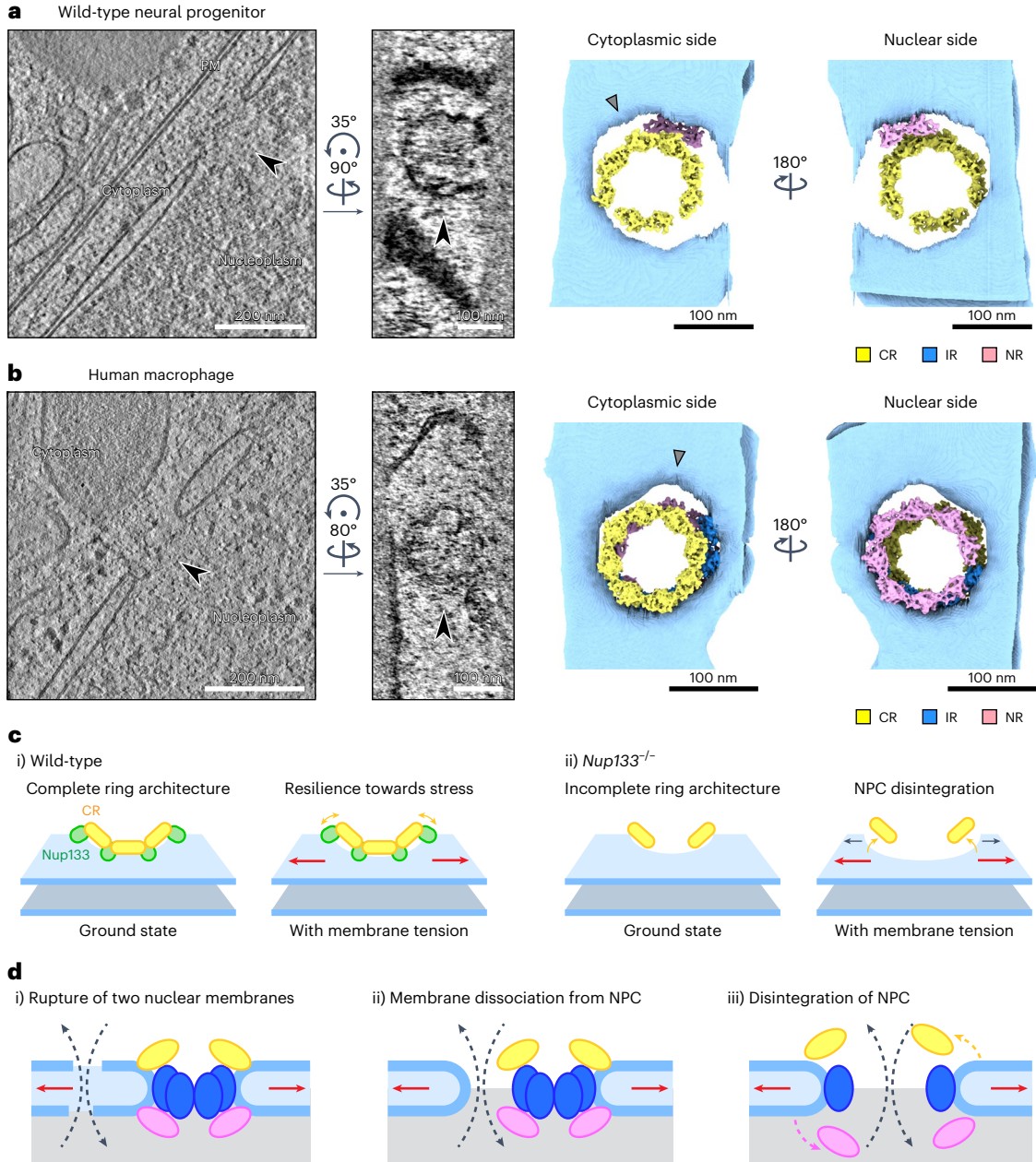

**Fig. 8 | Over-stretching of NPCs can occur in wild-type neural progenitor cells and human macrophages. a**, Representative example of an over-stretched NPC observed in the wild-type neural progenitor dataset. **b**, Representative example of an over-stretched NPC observed in the human macrophage dataset. In **a** and **b**, slices from the reconstructed tomograms and the pseudo-composite map generated from the results of the TM analysis are shown as in Fig. 6b. Sites of CR detachment from the outer nuclear membrane are highlighted with grey arrowheads. Note that no IR protomer is detected for the over-stretched NPC in the wild-type neural progenitor cells shown in **a**. All analysed over-stretched NPCs are shown in Extended Data Fig. 10. **c**, Model of NPC over-stretching in (i) wild-type and (ii) *Nup133*⁻/⁻ cells. In wild-type cells, Nup133 (depicted as green spheres) mediates the inter-protomer contacts and membrane anchoring of the CR (yellow) and NR (not depicted), thereby forming rigid ring architectures

that are resilient to membrane tension (red arrows) (i). In *Nup133*⁻/⁻ cells, the lack of Nup133 leads to unstable inter-protomer contacts and partial loss of CR protomers, making the CR architecture structurally less robust and more sensitive to membrane tension (ii). Yellow arrows indicate the motion of the NPC scaffolds, and dark grey arrows indicate the motion of the fusion point of the outer and inner nuclear membranes. **d**, Cartoons depicting possible mechanisms of nuclear envelope rupture. Nuclear envelope rupture could occur either by tearing of both the outer and inner nuclear membranes (i), by detachment of the nuclear envelope from the intact NPC scaffold architecture (ii), or by disintegration of the NPC (iii). The over-stretched NPCs observed in this study support the scenario depicted in (iii). Black dashed arrows indicate the leakage of cytoplasmic and nuclear contents. Red arrows depict increased nuclear envelope membrane tension. The rings are coloured as in Fig. 6a.

the intriguing possibility that the CR and NR protomers can be heterogeneously present within one NPC particle. This notion challenges the canonical assumption that all the symmetrically arranged protomers within NPCs are homogeneously present and can thus be treated equally during averaging-based structural analyses. Indeed, the NPC structural heterogeneity in *Nup133*⁻/⁻ cells was not detectable in the

initial subtomogram averages of the eight-fold symmetric NPCs (Figs. 1e and 3i), highlighting the necessity of complementary approaches, such as TM analysis, that do not require particle averaging. A heterogeneous presence of CR and NR protomers would be particularly relevant when the NPC architecture is perturbed experimentally, as in the *Nup133*⁻/⁻ cells, or deteriorated as an outcome of pathological conditions[59].

In such conditions, it is also possible that only a subset of particles possess severe architectural defects, like the over-stretched NPCs within our dataset. The TM analysis could be beneficial for analysing these minor populations of particles.

One further implication of our findings is that NPCs could be the sites where nuclear envelope rupture occurs. Previous studies have extensively investigated nuclear envelope rupture using cell biological approaches[43,45,46,60,61], and have established certain hallmarks and indicators of a rupture event, such as the detection of cytosolic DNA by cyclic GMP-AMP synthase (cGAS), or the accumulation of nuclear DNA damage. Due to the fact that nuclear envelope rupture events are rare and transient, it remains technically challenging to firmly establish that the NPC disintegration events we observed in our tomograms spatiotemporally coincide with cell biological hallmarks of rupture. We thus acknowledge that our experimental results provide only correlative evidence. Nevertheless, our data indicate the need to revisit the prevailing hypothesis that nuclear envelope rupture occurs by the simultaneous tearing of both outer and inner nuclear membranes (Fig. 8d,i). Given the observations in our cryo-electron tomograms, two possible additional scenarios have to be considered: the nuclear membranes remain unaffected and detach from the intact NPC scaffold at the fusion point of the nuclear membranes[10] (Fig. 8d,ii) or the NPC scaffold disintegrates and generates a larger membrane opening (Fig. 8d,iii).

Multiple possibilities have been proposed as underlying mechanisms of the *Nup133*⁻/⁻ cell differentiation defect, such as impaired nuclear basket assembly[20] or altered gene regulation[22]. Our findings now point to the additional possibility that impaired nuclear envelope integrity due to the disintegration of aberrant NPCs may impact cell differentiation. Intriguingly, a similar neuronal differentiation defect was also observed in mES cell lines devoid of the Y-complex components Nup43 and Seh1[21], supporting our notion that the defect is probably attributable to an impaired NPC architecture, rather than specific functions of individual Nups. We speculate that, because the size of the over-stretched NPC is larger and more variable in the *Nup133*⁻/⁻ neural progenitor cells (Fig. 5d), the NPC disintegration and potential damage to nuclear envelope integrity may be exacerbated during cell differentiation and cells may fail to properly differentiate when damage levels exceed the permissible degree. In addition to directly impairing nuclear envelope integrity, over-stretched NPCs might release the tension transmitted from the cytoskeleton networks, and thereby possibly perturb nuclear mechanosensing, which plays a key role in cell differentiation[34]. Although the exact contribution of NPC over-stretching and disintegration to the differentiation defect awaits further investigation, the postulated requirement of an intact NPC scaffold in a physiological context is further supported by its link to human genetic diseases. Specifically, mutations in the *Nup133* gene cause steroid-resistant nephrotic syndrome (SRNS)[14] and Galloway-Mowat syndrome[17], which involves neurological abnormalities in addition to the nephrotic syndrome. Intriguingly, some of the disease-causing mutations in Nup133 were demonstrated to disrupt its interaction with Nup107[14,17], and, conversely, a causal mutation of SRNS in the *Nup107* gene is known to disturb its interaction with Nup133[12,14]. Moreover, causal mutations of SRNS have also been found in two other Y-complex Nups, Nup160 and Nup85[14], and also in Nup93 and Nup205[62], both of which are involved in the head-to-tail contact between the Y-complexes (Fig. 2). Overall, these findings underline the physiological importance of an intact NPC scaffold architecture.

In summary, our study illustrates that perturbation of the NPC scaffold impacts the structural completeness and stability of the NPC in a heterogeneous manner, rather than causing homogeneous and large-scale structural impairment, and thereby provides insights into how mutations in Nups could cause various and complex phenotypes in cells. Moreover, this study also uncovers a critical role of the NPC scaffold in protecting membrane openings from excess expansion, whereby the NPC scaffold may be conceived as an annular spring with elastic properties that could act to safeguard the nuclear envelope. We envision that this safeguarding function of the NPC should be relevant for various biological processes that impose mechanical stress on the nucleus, such as cell adhesion, differentiation or migration.

## Online content

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

## Methods

Cell culture experiments were conducted using previously established mES cell lines, and no animal specimens were used. The cryo-ET data of macrophages have been reported in a previous study[53], where samples were prepared according to the regulations of the local ethics committee at Heidelberg University Hospital.

### mES cell culture and neural progenitor differentiation

For mES cell culture, the wild-type HM1 line[63] and the HM1-derived *Nup133*[−/−] clone (#14) described previously[20] were used. Mouse embryonic stem cell medium (mES medium) was prepared by supplementing EmbryoMax DMEM (Millipore, SLM-220-B) with 15% fetal bovine serum (FBS; Gibco, 10270-106), 2 mM L-glutamine (Gibco, 25030-024), 1× non-essential amino acids (Gibco, 11140-035 or 11140-050), 1× EmbryoMax nucleosides (Millipore, ES-008-D), 110 μM β-mercaptoethanol (Gibco, 21985-023) and 1× penicillin/streptomycin (Gibco, 15070-063). mES cells were cultured on mitomycin-treated mouse embryonic fibroblasts (feeder cells) plated on 0.1% gelatin (Sigma-Aldrich) in mES medium supplemented with leukaemia inhibitory factor (LIF, ESGRO, Millipore, final 1,000 U ml$^{-1}$), and kept at 37 °C with 5% $CO_2$. Cells were used within eight passages to ensure the quality of the culture.

The neural progenitor differentiation was performed following a previously published protocol[38]. Briefly, after two passages on feeder cells, mES cells were transferred to gelatin-coated plates without feeder cells, and cultured for two to four passages in the presence of LIF. Subsequently, the cells were trypsinized and $4 × 10^6$ cells were transferred to non-coated 10-cm Petri dishes (Greiner, 633102) and cultured for eight days in Dulbecco's modified Eagle medium (DMEM) supplemented with 10% FBS (Gibco, 10270-106), 2 mM L-glutamine (Gibco, 25030-024), 1× minimum essential medium (MEM) non-essential amino acids (Gibco, 11140-035) and 100 μM β-mercaptoethanol (Gibco, 31350-010) without LIF. The culture medium was changed every two days, and, after four days, 5 μM retinoic acid (Sigma, R2625) was added to the medium. After eight days of culture, the aggregates of neural progenitors were dissociated by trypsinization at 37 °C for 5 min in a water bath, and the dissociated cells were either immediately used for cryo-EM grid preparation as described below, frozen within the medium supplemented with 10% dimethyl sulfoxide (DMSO) and stored until use, or directly used for subsequent neuronal differentiation. For the neuronal differentiation, dissociated cells were seeded on glass-bottom plates (ibidi, 80827) coated with poly-DL-ornithine (Sigma, P8638) and laminin (Roche, 11243217001) at -1–2 × 10$^5$ cells cm$^{-2}$, and cultured in N2 medium (1:1 mixture of DMEM (Gibco, 21969-035) and Ham's F-12 nutrient mix (Gibco, 21765-029), supplemented with 2 mM L-glutamine, 25 μg ml$^{-1}$ insulin (Sigma, I6634), 50 μg ml$^{-1}$ transferrin (Sigma, T1147), 20 nM progesterone (Sigma, P8783), 100 nM putrescine (Sigma, P5780), 30 nM sodium-selenite (Sigma, 214485), 50 μg ml$^{-1}$ bovine serum albumin (BSA; Sigma, A9418) and 1× penicillin/streptomycin (Gibco, 15140-122)) for two days. The medium was changed to fresh N2 medium 2 h and one day after plating. Images of the cells were acquired on a Nikon Ti-2 widefield microscope (Nikon), using a ×10 objective.

### Cryo-EM sample preparation

For the grid preparation of mES samples, the cells were treated with a double thymidine block and synchronized at the beginning of S-phase to obtain a homogeneous cell population. Specifically, mES cells were cultured within mES medium supplemented with LIF for one day after passaging, and subsequently cultured within medium containing 2 mM thymidine (Sigma, T9250) for 18 h. The cells were then washed and cultured without thymidine for 9 h. After this short release from the cell-cycle arrest, the cells were again cultured in the presence of 2 mM thymidine for 18 h. The arrested cells were trypsinized and resuspended in the fresh medium at a density of 2–4 × 10$^6$ cells ml$^{-1}$ for subsequent use.

Cryo-EM grid preparation was performed using a EM GP2 plunger (Leica Microsystems), with a set chamber humidity of 70% and temperature of 37 °C. After trypsinization of the mES cells, 4 μl of the cell suspension was applied onto a glow-discharged holey SiO$_2$ grid (R1/4, Au, 200 mesh, Quantifoil) mounted on the plunger chamber, and 2 μl of the culture medium was applied onto the back side of the grid for reproducible blotting. Subsequently, the grid was blotted from the back side for 10 s with Whatman filter paper, grade 1, and plunge-frozen in liquid ethane. For the neural progenitor sample, the dissociated cells from the cell aggregate formed after the eight-day culture within non-coated dishes were plunge-frozen using the same procedure, but without a double thymidine block.

Plunge-frozen samples were cryo-FIB-milled using an Aquilos microscope (Thermo Scientific). The loaded samples were first coated with a layer of organometallic platinum using a gas injection system for 10 s, and subsequently sputter-coated with inorganic platinum at 1-kV voltage and 10-mA current for 20 s. Milling was performed in a stepwise manner, with decreasing Ga ion beam current steps from 1 nA to 30 pA at a stage tilt angle of 15°. Micro-expansion joints[64] were generated using an ion beam current of 1 nA or 500 pA alongside the lamellae to prevent lamella bending. Final polishing was performed using a 30-pA ion beam current, with a target lamella thickness of 180 nm.

### Cryo-ET data acquisition and tomogram reconstruction

Cryo-ET data acquisition was performed on a Titan Krios G2 microscope (Thermo Scientific) equipped with a BioQuantum K3 imaging filter (Gatan), operated at 300 kV in EFTEM mode with an energy slit width of 20 eV. Tilt series were acquired using SerialEM[65] in low dose mode at a magnification of ×33,000, corresponding to a nominal pixel size of 2.682 Å, with a defocus range of −2.5 to −4.5 μm. Images were recorded as movies of ten frames with a total dose of 2.5 e$^-$/Å$^2$, and motion-corrected in SerialEM on-the-fly. Tilt series acquisition started from the stage pretilt angle of 8° to compensate for the lamella angle, and images were acquired using a dose-symmetric acquisition scheme[66] with 2° increments, up to +64° or 66° and −52° stage tilt angle, resulting in a total dose per tomogram of -150 e$^-$/Å$^2$.

Before tomographic reconstruction, the contrast transfer function (CTF) for each tilt image was estimated using gctf[67]. Subsequently, images were filtered by cumulative electron dose[68], using a MATLAB implementation as described previously[69]. Next, tilt images with poor quality were discarded by visual inspection. These dose-filtered and cleaned tilt series were aligned with patch-tracking in IMOD (version 4.10.9 and 4.11.5)[70], and reconstructed as 4× binned, SIRT-like filtered tomograms with a binned pixel size of 10.728 Å. Tomograms with sufficient visual quality and a good patch-tracking result (typically, an overall residual error value below 1 pixel at 4× binning) were selected and 3D CTF-corrected using novaCTF[71], then used for the subtomogram averaging workflow. For the tomogram-based NPC diameter inspection, all reconstructed tomograms were visually examined. Only tomograms with a severe defect, such as overall residual error value above three pixels, total angular coverage less than 60° or a large amount of ice contamination, were discarded, then the remaining tomograms were used for analysis. For visualization of the tomographic slices, 4× binned 3D CTF-corrected tomograms were deconvolved in MATLAB using the tom_deconv deconvolution filter[72] to enhance the image contrast.

### Subtomogram averaging and classification

Subtomogram averaging of NPCs was performed using novaSTA[73] as described previously[10] (the detailed workflows are summarized in Extended Data Fig. 3a). Briefly, NPC particles were manually picked using 4× binned SIRT-like filtered tomograms, extracted from 8× binned 3D CTF-corrected tomograms, and aligned on a whole-pore level with imposed C8 symmetry using novaSTA[73]. After initial alignment at 8× binning, the particles were again extracted from

4× binned 3D CTF-corrected tomograms and further aligned to improve the alignment precision. After obtaining the whole-pore average, the positions of asymmetric units were determined using the distance from the centre of the whole pore and C8 symmetry, and subtomograms for the asymmetric units were extracted at 4× binning. Particles with the centre outside lamellae were discarded at this step. Subsequently, subtomograms were aligned, and particles with a cross-correlation value to the reference below 0.1 were further discarded. To obtain isotropic averages, subtomograms from the top views of NPCs were also discarded from the dataset at this step, and the remaining particles were further processed. For the subtomogram average-based NPC diameter measurement, particles from top views were kept, and the entire particle set was processed in parallel. After averaging the asymmetric unit, the alignment around each ring subunit (that is, CR, IR and NR) was further optimized using a mask covering these target areas. Subsequently, subtomograms were re-extracted from the centre of individual ring subunits at 4× binning, and further aligned to obtain final averages. The masks used for the alignment were generated with Dynamo[74] and RELION-3.1[75]. To generate the composite maps of the whole pore, the final individual ring averages were fitted into the average of the asymmetric unit, and the summed map was assembled into an eight-fold symmetric architecture based on the coordinates used for asymmetric unit extraction.

For the *Nup133*−/− mES dataset, the whole-pore particles were subjected to reference-based classification using STOPGAP[76], to classify particles with seven-, eight- and nine-fold symmetries. To generate initial references with three different symmetries, the particles of *Nup133*−/− mES NPCs were first manually sorted based on the arrangement of asymmetric units. Specifically, particles were initially picked from the tomograms without symmetry judgement and subjected to whole-pore averaging at 8× binning with imposed C8 symmetry. Subsequently, asymmetric units were extracted using C8 symmetry and aligned as described above to obtain the initial asymmetric unit average. In parallel, asymmetric units were extracted using C60 symmetry to oversample the positions on the circumference of the NPCs, and aligned using the above-mentioned asymmetric unit average as the initial reference. After 16 cycles of iterative alignments, most particles were shifted to the actual positions of the closest subunit, and these shifted positions of the oversampled particles were visually examined to assess the symmetry of each NPC. As a result, 23, 261 and 18 NPC particles out of 411 particles showed clear seven-, eight- and nine-fold symmetric arrangement of the asymmetric units, respectively, and these particles were aligned with C7, C8 or C9 symmetry at 8× binning to obtain three initial references for the classification. With these whole-pore averages with seven-, eight- and nine-fold symmetries as initial references, the NPC particles were subsequently subjected to reference-based classification using STOPGAP[76] without any imposed symmetry. Among the 411 initially picked NPC particles, 11 particles at the very edge of the tomograms contained empty areas within their subtomograms, and thus were discarded from the particle set before the classification analysis. Because STOPGAP uses stochastic methods for subtomogram alignment[76], classification was repeated four times using the same parameters, and particles consistently classified into the same classes in all four runs were selected for further processing. The particles with eight-fold symmetry were processed as described above, whereas those with seven-fold or nine-fold symmetry were only processed at 8× binning in a similar manner. For the *Nup133*−/− neural progenitor dataset, 193 whole-pore particles were extracted and directly subjected to the reference-based classification, using the three initial references used for the classification of the *Nup133*−/− mES NPCs. Due to the low number of particles classified into the seven-fold symmetric class and the presence of misclassified particles, final averages for the seven-fold symmetric class lacked readily distinguishable symmetric features in some of the classification runs. Thus, to reliably separate seven-fold and eight-fold symmetric particles, classification was performed eight times, and five runs that yielded clear seven-fold and eight-fold symmetric averages were used for selecting the consistently classified particles. No particle was classified as a nine-fold symmetric NPC in any of the classification runs. One-hundred and fifty seven particles consistently classified as eight-fold symmetric NPCs were further processed as described above to obtain final averages of the CR, IR and NR. The numbers of particles used for individual subtomogram averages are summarized in Supplementary Table 1.

Subtomogram average-based NPC diameter measurements were performed as described previously[24]. For the manual measurement of NPC diameter, 4× binned SIRT-like filtered tomograms were visually examined, and the distance between two opposite nuclear envelopes at the outer–inner nuclear envelope fusion point was measured using the distance measurement function in 3dmod[70]. When NPCs exhibited an ellipsoidal or distorted shape, the measurement was performed at the widest section. Figures were prepared using UCSF ChimeraX[77].

## Human macrophage dataset

The cryo-ET data of macrophages were acquired in a separate study[53]. In brief, monocyte-derived macrophages (MDMs) were obtained from human peripheral blood mononuclear cells, which were isolated from buffy coats of anonymous blood donors at the Heidelberg University Hospital Blood Bank by Ficoll density gradient centrifugation[78], according to the regulations of the local ethics committee. Isolated MDM cells were cultured in RPMI 1640 medium (Thermo Fisher Scientific) supplemented with 10% heat-inactivated FBS (Capricorn Scientific), 100 U ml−1 of penicillin, 100 μg ml−1 of streptomycin (Thermo Fisher Scientific) and 5% human AB serum (Capricorn Scientific) at 37 °C in a humidified incubator with a 5% $CO_2$ atmosphere. For grid preparation, MDM cells were detached by accutase (StemCell Technologies) according to the manufacturer's instructions, seeded on the grids, and incubated for an additional 24 h at 37 °C before plunge freezing. Plunge freezing and cryo-FIB milling of the samples were performed in a similar manner as described above for mES and neural progenitor samples. Tomogram reconstruction was performed as described above, but AreTomo (version 1.33)[79] instead of IMOD was used for patch-tracking.

## Three-dimensional template matching

The 3D TM was performed using STOPGAP[76]. To reduce the amount of computation, selected whole-pore particles were extracted from 4× binned 3D CTF-corrected tomograms with a box size of 240 or 300 voxels and subjected to analysis. For analysis of the eight-fold symmetric NPCs, particles located within the central 616 × 1,040 × ~200-voxel volume of 4× binned tomograms (1,016 × 1,440 × 500- or 600-voxel total volume) were first selected to exclude particles close to the edge of the tomograms and thus that might have been partially out of the volume. From these particles, 40 and 30 particles for the wild-type and *Nup133*−/− mES datasets, respectively, were randomly selected to have particle sets of similar size to those of the seven-fold and nine-fold symmetric NPCs. For analysis of the seven-fold and nine-fold symmetric NPCs, all the particles obtained from reference-based classification were used. The individual ring averages (CR, IR or NR) of the eight-fold symmetric NPCs from the wild-type or *Nup133*−/− mES cells, aligned at 4× binning with a box size of 100 voxels, were directly used as search templates. Ellipsoidal masks that cover the protein part of the averages were used as alignment masks. Low-pass filtering of ~50 Å was applied to the templates based on the resolution of the final averages. An angular search was performed with 10° increments without any symmetry. After template matching runs, the mean and s.d. values of the cross-correlation scores within each score volume were calculated[42], and the cross-correlation peaks with values larger than 5 s.d. from the mean were extracted using sg_tm_generate_motl.m in the STOPGAP toolbox[76]. The extracted peaks were inspected using ArtiaX[80], and only the peaks with consistent particle orientations to the NPC architectures

were kept. The positions of the peaks were further visually examined together with the corresponding cross-correlation score volumes, and peaks indistinguishable from background noise were discarded. For the analysis of the number of CR and NR peaks shown in Figs. 3 and 4, particles with fewer than five IR peaks were also discarded, because the overall symmetry of the NPC (seven-, eight- or nine-fold symmetry) cannot be unambiguously judged from the arrangement of the IR peaks. Subvolumes used for TM analysis were then visually inspected, and particles that were not fully contained within the lamellae were excluded. Because the seven-fold and nine-fold symmetric NPC particles obtained after classification nevertheless included a small number of misclassified NPCs or particles with ambiguous symmetries, these particles were also discarded based on the arrangement of the IR peaks. As a result, 10 out of 34 particles, 6 out of 30 particles, 12 out of 32 particles and 11 out of 40 particles were discarded from the datasets of the $Nup133^{-/-}$ mES seven-fold, eight-fold, nine-fold symmetric NPCs and the wild-type mES NPCs, respectively. For the human macrophage dataset, selected particles were extracted from 4× binned 3D CTF-corrected tomograms (9.656 Å/pixel) with a box size of 300 voxels. The CR, IR and NR averages of the wild-type mES NPCs were rescaled to the corresponding pixel size using relion_image_handler[75], together with the corresponding masks, and used as search templates. Template matching was performed as described above, and the cross-correlation peaks with value larger than 6 s.d. above the mean were extracted and analysed. Nuclear envelopes were segmented from deconvolved tomograms[72] using Membrain-seg[81] with pretrained model version 10, and the results were corrected manually using Amira Software (Thermo Scientific). For visualization of the results, MATLAB, ArtiaX[80] and napari[82] were used.

## Immunofluorescence staining

For mES samples, feeder cells were first seeded on eight-well polymer-bottom chambered slides (ibidi, 80826) coated with 0.1% gelatin at a seeding density of $2 \times 10^4$ cells cm$^{-2}$. mES cells were seeded on feeder cells at a density of $4 \times 10^4$ cells cm$^{-2}$, and cultured for 24 h before fixation with 4% paraformaldehyde for 15 min. For neural progenitor samples, cells were seeded on eight-well glass-bottom chambered slides (ibidi, 80827) coated with poly-LD-ornithine and laminin as described above, and fixed at the indicated time point after seeding in 4% paraformaldehyde for 15 min. For staining, cells were permeabilized in 0.1% Triton in phosphate buffered saline (PBS) for 10 min, and blocked for 30 min using 3% BSA-containing PBS (PBS/BSA). After blocking, primary antibodies diluted in PBS/BSA were applied, and the samples were incubated overnight at 4 °C. Subsequently, the samples were incubated with secondary antibodies diluted in PBS for 2 h at room temperature. DNA was stained with 1 μM Hoechst 33342 (Thermo Fisher Scientific, 62249). Images were acquired on a Stellaris 5 confocal microscope (Leica Microsystems), using a ×63/1.40 oil objective. For quantification of Pax6-positive cells, 15 $z$-slices were acquired with a $z$-step of 1 μm, and 25 $z$-slices were acquired with a $z$-step of 0.6 μm for the analysis of DNA damage foci and nuclear morphology. Typically, images were acquired from 8 to 12 different positions within one well, and all the images were used for the analysis. The primary antibodies used in this study were as follows; mouse anti-Pax6 (Developmental Studies Hybridoma Bank, RRID AB_528427, 61 μg ml$^{-1}$, dilution 1:12), mouse anti-Oct3/4 (Santa Cruz Biotechnology, sc-5279, 200 μg ml$^{-1}$, dilution 1:200), mouse anti-γ-H2AX (Merck, 05-636, 1 mg ml$^{-1}$, dilution 1:1,000), rabbit anti-53BP1 (Novus Biologicals, NB100-304SS, 1 mg ml$^{-1}$, dilution 1:1,000) and rabbit anti-YAP (Cell Signaling Technology, #14074, 7 μg ml$^{-1}$, dilution 1:100). For the secondary antibodies, goat anti-mouse Alexa Fluor Plus 488-conjugated antibody (Thermo Fisher Scientific, A32723, 2 mg ml$^{-1}$, dilution 1:2,000) and goat anti-rabbit Alexa Fluor 594-conjugated antibody (Thermo Fisher Scientific, A11012, 2 mg ml$^{-1}$, dilution 1:2,000) were used.

Image analysis was performed using Fiji[83]. For quantification of Pax6-positive cells, nuclei and Pax6-positive areas were segmented in maximum intensity projection images by auto-thresholding, and the percentage of Pax6-positive nuclei was calculated. For counting γ-H2AX and 53BP1 foci, nuclei were segmented in 3D using the 3D nuclei segmentation tool in the 3D ImageJ Suite[84], and γ-H2AX and 53BP1 foci were segmented from 3D median-filtered stacks using a defined threshold value. Threshold values were determined for each dataset based on an intensity histogram (top 0.02 percentile; the top 0.1 percentile was used for the analysis of γ-H2AX foci at the 2 h time point due to high signal intensities in the subset of the cells), and a single threshold value was used for all images from both wild-type and $Nup133^{-/-}$ samples. The number of foci larger than three pixels was counted for each nucleus using 3D Objects Counter[85]. The average YAP intensity in the nuclei was quantified using the 3D Intensity Measure tool in the 3D ImageJ Suite[84]. Nuclei with an average nuclear YAP intensity value higher than 3,000 were judged to be YAP-positive. For quantification of the nuclear morphology, the size of the bounding box of the segmented nuclei was obtained from 3D Objects Counter[85], and the ratio of its height and the shorter horizontal axis was used as a flatness index (height/width). For the neuronal precursor samples prepared at the 2 h time point, nuclei with a flatness index higher than 3 were discarded as outliers (two from each dataset).

## Cell viability assay against DNA-damaging agents

Cell viability in the presence of DNA-damaging agents was analysed by a CellTiter-Glo 2.0 Cell Viability Assay (Promega, #G9242). For the mES samples, $1 \times 10^4$ cells were seeded into 96-well clear-bottom black plates (Corning, #3603) coated with 0.1% gelatin, and were cultured for 24 h in mES medium supplemented with LIF. Subsequently, DNA-damaging agents were added to the wells at the indicated final concentrations, and the cells were cultured for 48 h. For the neural progenitor samples, $4–5 \times 10^4$ cells were seeded into 96-well clear-bottom black plates (Corning, #3603) coated with poly-DL-ornithine and laminin, and were cultured for 6 h in N2 medium. Subsequently, DNA-damaging agents were added to the wells at the indicated final concentrations, and the cells were cultured for 18 h. After collecting the plates, 100 μl of CellTiter-Glo 2.0 reagent was added to each well, and the plates were incubated at room temperature for 10 min before luminescence measurements. Luminescence was measured using a Tecan Spark microplate reader, and the data were normalized against the viability of the mock-added cells to derive the relative viability percentage. Cisplatin (Merck, 232120), hydroxyurea (Merck, H8627) and bleomycin (Selleckchem, S1214) were dissolved in PBS at a concentration of 3 mM, 100 mM and 50 mM, respectively, and etoposide (Merck, 341205) was dissolved in DMSO at a concentration of 50 mM. Serial dilutions were prepared in PBS from these stock solutions and used for the assay. The data were plotted using GraphPad Prism 9.

## Statistics and reproducibility

Statistical analysis was performed using GraphPad Prism 9, and the statistical significance was analysed using an unpaired two-tailed $t$-test or a Kruskal–Wallis rank sum test followed by Dunn's multiple comparisons test. Where statistical comparisons were performed, specific $P$ values are indicated on the graphs. No statistical method was used to predetermine sample size. Sample sizes were determined based on similar experiments reported in previous studies and in consideration of data acquisition parameters. All experiments were repeated two to four times using samples from independent sample preparation or cell differentiation sessions. The exact sample size and number of biological replicates are specified in the respective figure legends. No data were excluded from the analyses. The experiments were not randomized. The investigators were not blinded to allocation during the experiments and outcome assessment due to the obvious phenotypes present in the microscopy images. The data distribution was assumed to be normal but this was not formally tested.

## Reporting summary

Further information on research design is available in the Nature Portfolio Reporting Summary linked to this Article.

## Data availability

The subtomogram averages described in this study are deposited in the Electron Microscopy Data Bank (EMDB) with accession codes EMD-52153, EMD-52154, EMD-52155, EMD-52156, EMD-52157, EMD-52158, EMD-52159, EMD-52160, EMD-52161, EMD-52162, EMD-52163 and EMD-52164. The composite maps of the *C*8-symmetrized whole NPCs are deposited as supplementary maps of EMD-52153, EMD-52156, EMD-52159 and EMD-52162. The raw tilt series are deposited in the Electron Microscopy Public Image Archive (EMPIAR) with accession codes EMPIAR-12457, EMPIAR-12460, EMPIAR-12461, EMPIAR-12462 and EMPIAR-12463. Light microscopy images used for quantitative analyses are deposited in the BioImage Archive with the accession codes S-BIAD1493, S-BIAD1494 and S-BIAD1495. Data supporting the findings of this study are available from the corresponding author on reasonable request. Source data are provided with this paper.

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

## Acknowledgements

We thank A. Becker, E. Kaindl and V. Pintschovius for technical support, and all the members of the Beck laboratory for discussion and advice. We thank A. Schwarz from the Max Planck Institute for Brain Research for technical advice. We thank S. Welsch, M. Linder, S. Prinz and S. Kaltwasser from the Central Electron Microscopy facility of the Max Planck Institute of Biophysics for assistance with cryo-EM sample preparation and data acquisition. We thank Ö. Yildiz, J. F. Castillo Hernandez, T. Hoffmann and the Max Planck Computing and Data Facility for computational resources. We acknowledge support from the Imaging Facility of the Max Planck Institute for Brain Research. We thank A.-M. Heuser, M. Anders-Ößwein and V. Sonntag-Buck for preparation of the human macrophage samples, and E. Margiotta for establishing the cryo-EM sample preparation workflow at the early stage of the project. M.B. acknowledges funding by the Max Planck Society. This research was supported by the German Research Foundation (CRC 1507 – Membrane-associated Protein Assemblies, Machineries and Supercomplexes (project no. 450648163); Project 17 to M.B. and CRC 1129 – Integrative Analysis of Pathogen Replication and Spread (project no. 240245660); Projects 5 to H.-G.K. and 20 to M.B.). This work was also supported by the European Union (ERC, NPCvalve, project no. 101054823 to M.B.). Views and opinions expressed are, however, those of the authors only and do not necessarily reflect those of the European Union or the European Research Council Executive Agency. Neither the European Union nor the granting authority can be held responsible for them. R.T. was supported by an EMBO long-term fellowship (ALTF 170-2019) and the Osamu Hayaishi Memorial Scholarship for Study Abroad from the Japanese Biochemical Society. V.D. acknowledges funding by the Centre National de la Recherche Scientifique (CNRS), the Fondation pour la Recherche Médicale (FRM, Foundation for Medical Research) under grant no. EQU202003010205 and by the Labex Who Am I? (ANR-11-LABX-0071; Idex ANR-11-IDEX-0005-02). C.O. received PhD fellowships from Ecole Doctorale BioSPC, Université Paris Cité and from the Fondation pour la Recherche Médicale (fourth year). The ImagoSeine core facility was supported by funds from the GIS-IBISA (groupement d'intérêt scientifique–Infrastructure en biologie santé et agronomie), the France-Bioimaging (ANR-10-INBS-04) infrastructures and la Ligue contre le cancer (RO3/75-79).

## Author contributions

R.T., C.O., V.D. and M.B. conceived the project. R.T. performed experiments with mES and neural progenitor cells and analysed data. C.O. and V.D. provided mES cell lines and supported R.T. with mES cell culture. J.P.K. performed cryo-FIB milling of human macrophages, and acquired and analysed data. V.Z. prepared cryo-EM samples of human macrophages under the supervision of H.-G.K. C.E.Z. supported R.T. with cryo-EM data analysis. S.B. supported R.T. with analysis of DNA damage foci. B.T. supported R.T. and J.P.K. with data analysis. R.T., S.B., V.D. and M.B. wrote the manuscript. V.D. and M.B. supervised the research.

## Funding

## Competing interests

The authors declare no competing interests.

## Additional information

**Extended data** is available for this paper at

**Supplementary information** The online version
contains supplementary material available at

**Correspondence and requests for materials** should be addressed to
Valérie Doye or Martin Beck.

**Peer review information** *Nature Cell Biology* thanks the anonymous
reviewers for their contribution to the peer review of this work.
Peer reviewer reports are available.

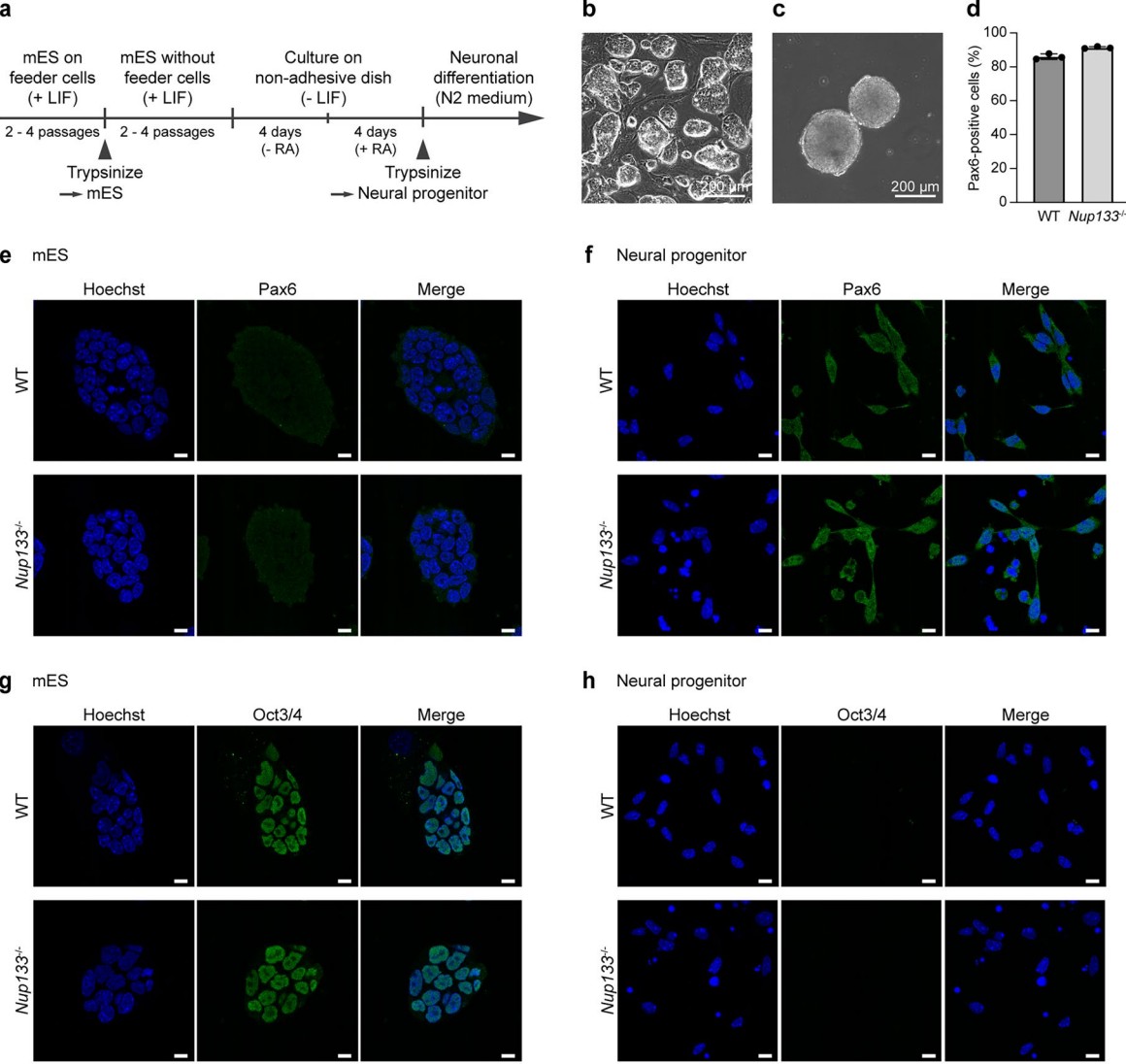

**Extended Data Fig. 1 | Overview of sample preparation for cryo-ET experiments.** (**a**) Flow diagram of the neuronal differentiation procedure. Time points when mES and neural progenitor cells were plunge-frozen are also indicated by arrowheads. (**b**) Representative image of the wild-type mES cells cultured on a layer of feeder cells. Oval colonies with bright edges are clusters of mES cells. (**c**) Representative image of a cluster of the wild-type neural progenitor cells. These clusters were trypsinized to obtain neural progenitor samples. (**d**) Mean percentage of Pax6-positive cells in the neural progenitor samples obtained from the wild-type and *Nup133⁻/⁻* mES cells. Cells were stained 2 h after plating with Pax6 antibody. Data are from three biological replicates, and at least 330 nuclei are analyzed for each replicate. Error bars denote standard deviations. (**e**, **f**) Representative immunofluorescence staining images of Pax6 in mES cells (**e**) and neural progenitor cells 2 h after neuronal differentiation induction (**f**). (**g**, **h**) Representative immunofluorescence staining images of Oct3/4 in mES cells (**g**) and neural progenitor cells 2 h after neuronal differentiation induction (**h**). In (**e-h**), single image frames from confocal stacks are shown. Experiments were repeated three times and consistent results were obtained. Scale bars, 10 μm. In (**e**, **g**), cells were fixed approximately 24 h after plating them onto feeder cells. In (**f**, **h**), cells were fixed at 2 h time point after induction of neuronal differentiation. Source numerical data are available in source data.

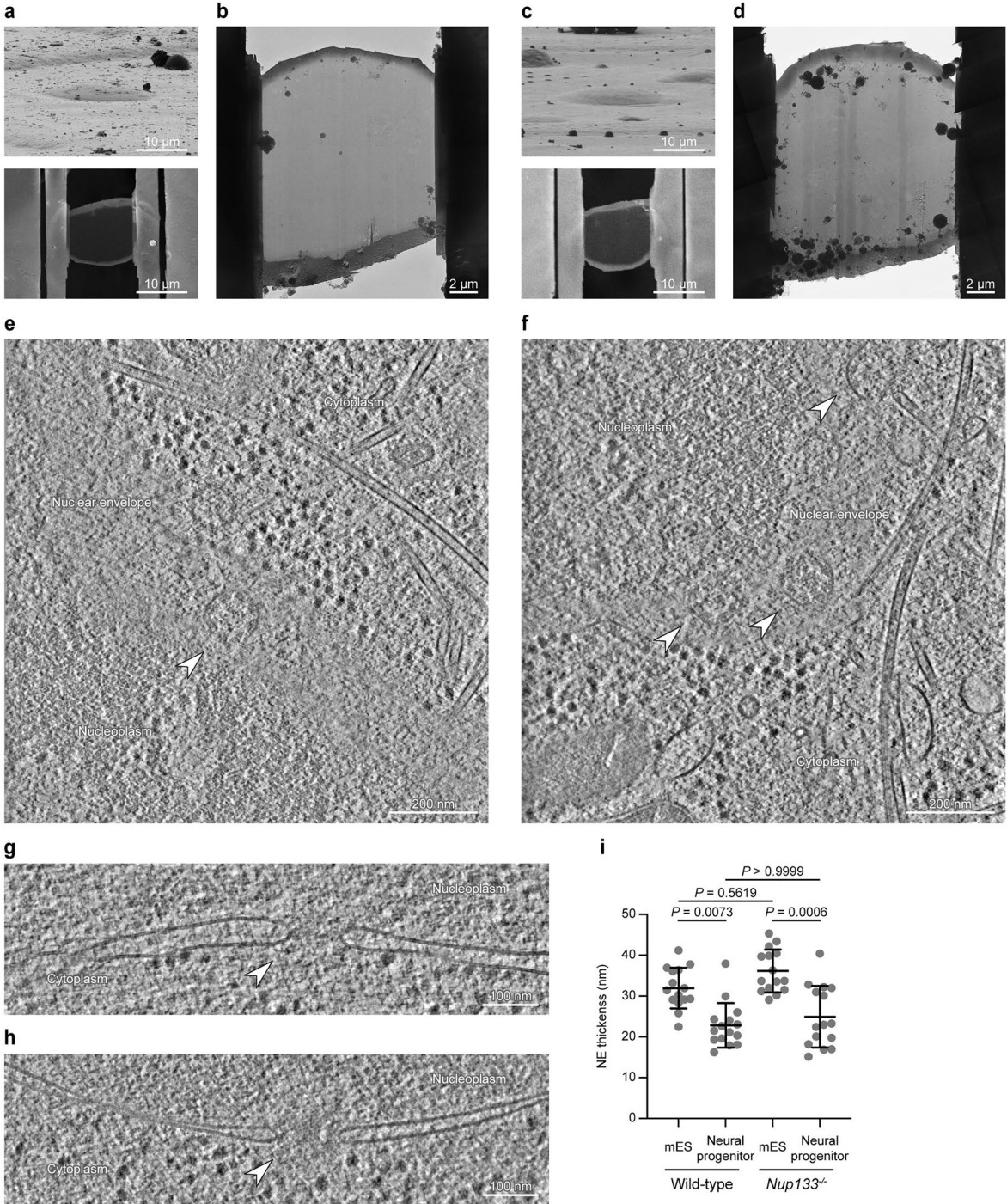

**Extended Data Fig. 2 | Overview of cryo-FIB milling and reconstructed tomograms.** (**a**) Representative cryo-FIB milled lamella of the wild-type mES cells. FIB view of an mES cell on a cryo-EM grid before cryo-FIB milling (top) and SEM view of the same cell after cryo-FIB milling (bottom) are shown. (**b**) Cryo-TEM overview of the lamella shown in (**a**), rotated by 180°. (**c**, **d**) Representative cryo-FIB milled lamella of the wild-type neural progenitor cells, shown as in (**a**) and (**b**). 41 and 27 lamellae of similar quality were prepared for the wild-type mES and neural progenitor datasets, respectively. (**e**, **f**) Representative slices through reconstructed tomograms of the wild-type mES cells (**e**) and the wild-type neural progenitor cells (**f**). Top views of the NPCs are indicated by white arrowheads. (**g**, **h**) Representative tomographic slices showing the side views of the nuclear envelope of the wild-type mES cells (**g**) and the wild-type neural progenitor cells (**h**). Side views of the NPCs are indicated by white arrowheads. For (**e** - **h**), the total numbers of acquired tomograms are provided in Supplementary Table 1. (**i**) Quantification of the nuclear envelope thickness. The thickness was measured manually at more than six representative positions within the tomograms, and the averaged values were plotted. Measurement was performed for randomly selected 15 tomograms with clear side views of the nuclear envelope. Bars denote the means and standard deviations. Kruskal-Wallis rank sum test followed by Dunn's multiple comparisons test. Source numerical data are available in source data.

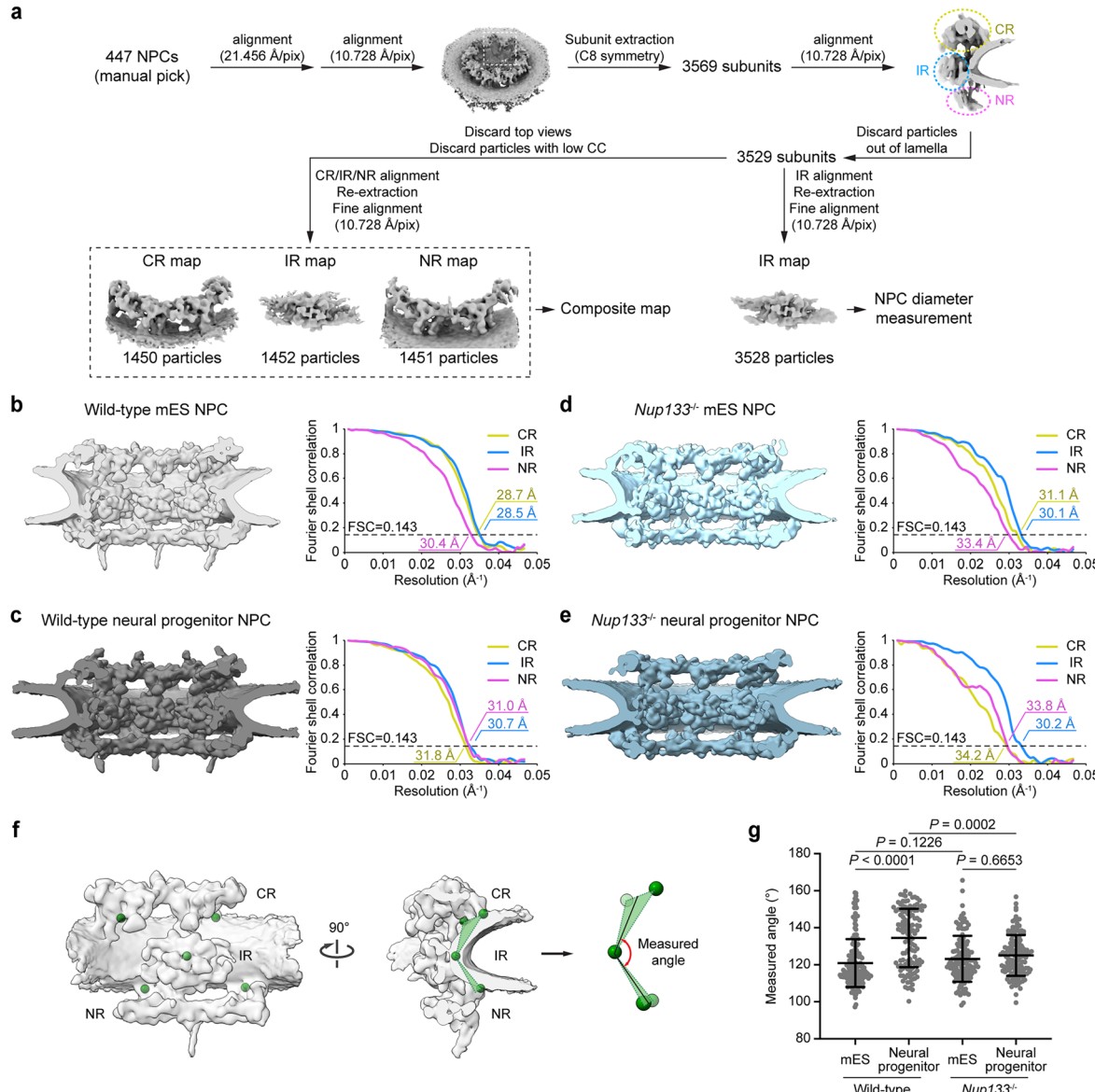

**Extended Data Fig. 3 | Subtomogram averages and structural comparison of the mES and neural progenitor NPCs.** (**a**) The schematic data processing workflow for subtomogram averaging of the wild-type mES NPCs. The eight-fold symmetric NPCs from the three other datasets were processed following a similar workflow. (**b** - **e**) Composite maps of the eight-fold symmetric NPCs (left) and corresponding Fourier shell correlation curves of the CR, IR and NR averages (right). The data for the wild-type mES NPC (**b**), the wild-type neural progenitor NPC (**c**), the *Nup133⁻/⁻* mES NPC (**d**), and the *Nup133⁻/⁻* neural progenitor NPC (**e**) are shown. Composite maps are shown as cutaway views. (**f**, **g**) Subtomogram-based measurement of the angles between the outer and inner nuclear membranes. In (**f**), the workflow of measurement is shown as a schematic representation. Based on the subtomogram averages of individual rings (CR, IR, and NR), five coordinates were derived for each NPC subunit

(membrane anchor sites of the outer Y-complex of CR and NR (two each from two adjacent protomers) and center of the membrane attachment interface of IR, depicted as green spheres). Subsequently, the angle between two planes (one formed between IR and two CR coordinates, the other formed between IR and two NR coordinates) were measured for each NPC subunit. The measured angles were averaged per NPC, and the averaged angles were plotted (**g**). NPCs with less than four measured values were discarded from the analysis. Bars denote the means and standard deviations (n = 184 NPCs for the wild-type mES dataset, n = 119 NPCs for the wild-type neural progenitor dataset, n = 128 NPCs for the *Nup133⁻/⁻* mES dataset, n = 136 NPCs for the *Nup133⁻/⁻* neural progenitor dataset). Kruskal-Wallis rank sum test followed by Dunn's multiple comparisons test. Source numerical data are available in source data.

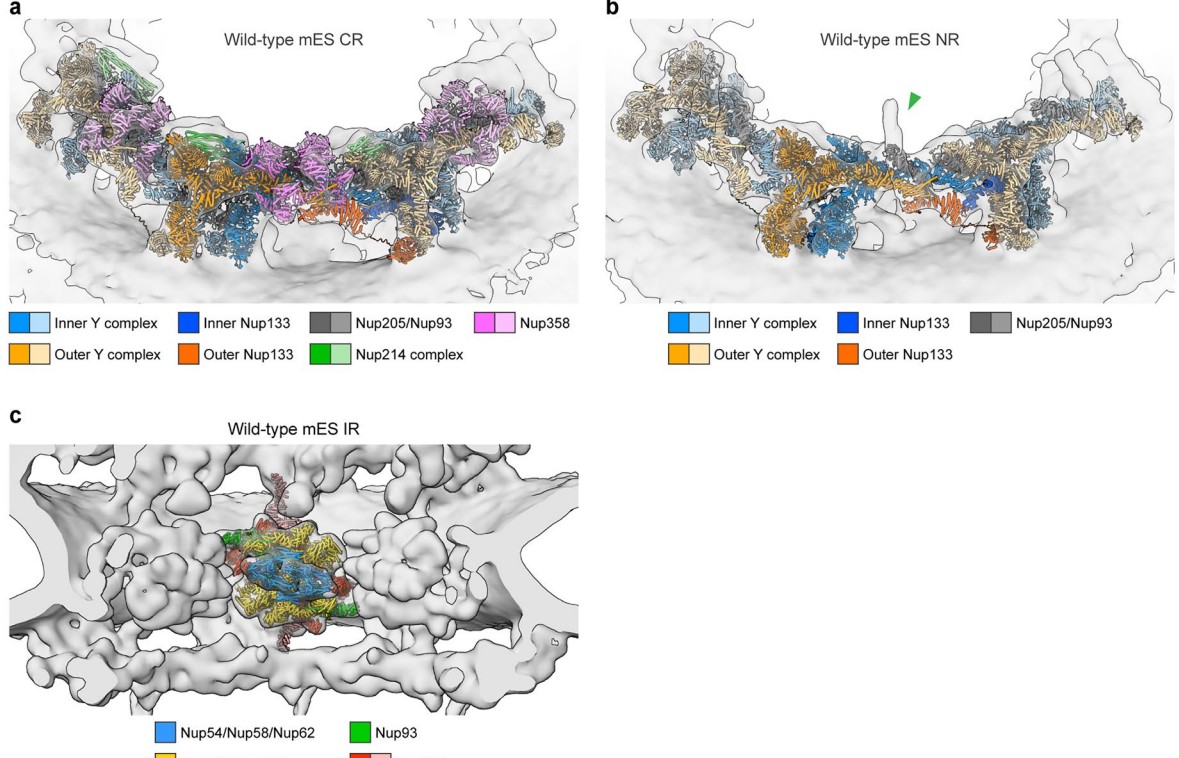

**Extended Data Fig. 4 | Structural comparison between murine and human NPCs. (a - c)** Detailed architecture of the CR (**a**), NR (**b**), and IR (**c**) of the wild-type mES NPCs. The CR, NR, and IR models of the dilated human NPC (PDBID: 7R5J) are fitted into the corresponding cryo-EM maps. In (**a**) and (**b**), protomers at the center are highlighted with bold colors, while two adjacent protomers are indicated with pale colors. Note that Nup358 and Nup214 complex are CR-specific components. In (**b**), the position of the basket filament is highlighted with a green arrowhead. In (**c**), Nup155 molecules linking the IR with the CR or NR are colored in pale red.

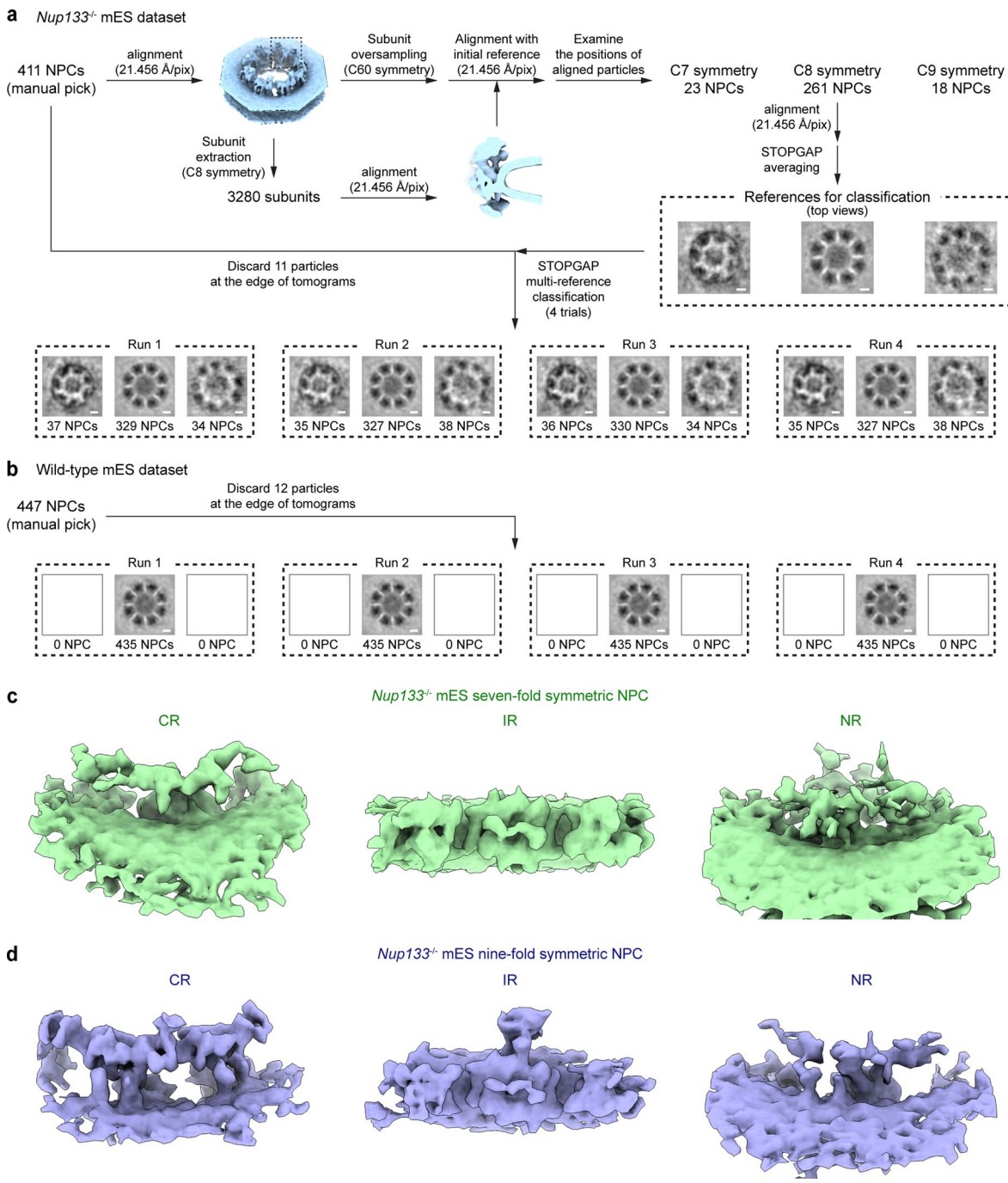

**Extended Data Fig. 5 | Subtomogram averaging of the seven-fold and nine-fold symmetric NPCs in the *Nup133*-/- mES cells.** (**a**) The schematic workflow of the classification of the *Nup133*-/- mES NPCs. Particles with consistent class assignment among four classification runs were selected for further processing. (**b**) Classification of the wild-type mES NPCs. The three initial references shown in (**a**) were used. In contrast to the classification of the *Nup133*-/- mES NPCs, all the particles were reproducibly assigned to one class with eight-fold symmetric architecture. Scale bars in (**a**) and (**b**), 20 nm. (**c**, **d**) The cryo-EM maps of the seven-fold (**c**) and nine-fold (**d**) symmetric NPCs in the *Nup133*-/- mES cells. The individual averages of the CR, IR, and NR are shown. The CR and NR averages are viewed from the same angle as in Fig. 1b, and the IR average is viewed from the center of the NPC. Note that the CR and NR averages in (**c**) and the NR average in (**d**) lack interpretable structural features.

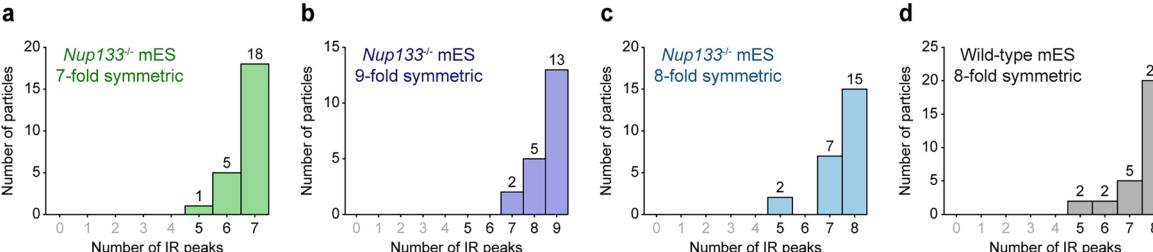

**Extended Data Fig. 6 | TM analysis of the mES NPCs. (a - d)** Histograms showing the number of NPCs and their differing numbers of IR peaks detected by the TM analysis of the seven-fold (**a**), nine-fold (**b**), and eight-fold (**c**) symmetric NPCs in the *Nup133-/-* mES cells, and the eight-fold symmetric NPCs in the wild-type mES cells (**d**). Note that particles with less than five peaks were discarded from the analysis (labels colored in grey). The number of particles is indicated on top of each bar. Graphs in (**a - d**) correspond to the dataset presented in Figs. 3c,e, 4b, and 4d, respectively.

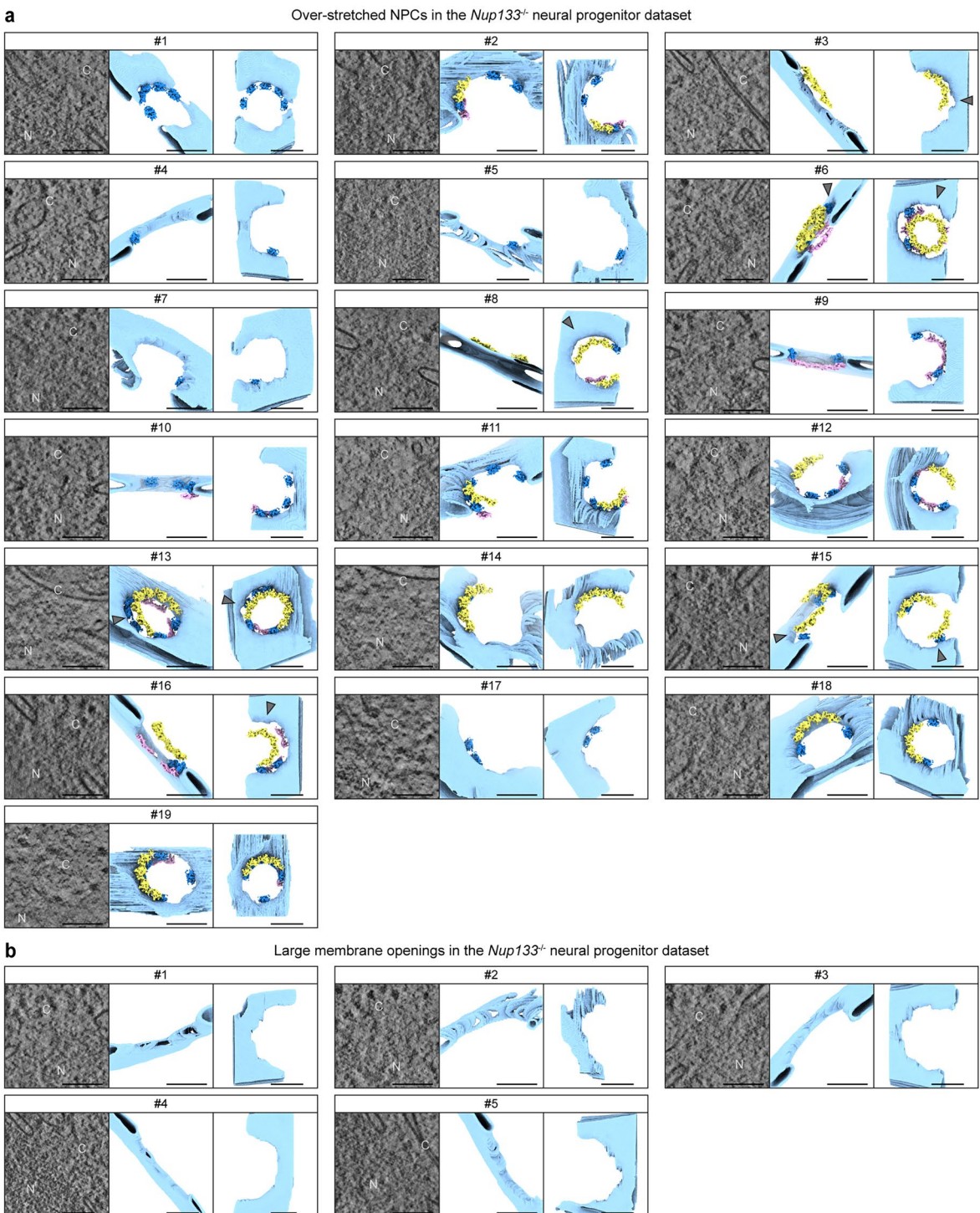

**Extended Data Fig. 7 | Over-stretched NPCs in the *Nup133*⁻ᐟ⁻ neural progenitor cells.** (**a**) Summary of the over-stretched NPCs observed in the *Nup133*⁻ᐟ⁻ neural progenitor dataset. Each example of an over-stretched NPC (#1–19) includes a representative slice through the tomograms (left), segmented membranes seen from the same orientation as in the left panel (middle), and segmented membranes seen from the cytoplasmic side (right). Based on the results of the TM analysis, the CR (yellow), IR (blue) and NR (pink) maps are projected back onto the segmented membranes (middle and right panels). The CRs clearly detached from the outer nuclear membrane are highlighted with grey arrowheads. C and N in the tomographic slices indicate the cytoplasm and nucleus, respectively. The example shown in Fig. 6b corresponds to #8. (**b**) Summary of large nuclear envelope openings observed in the *Nup133*⁻ᐟ⁻ neural progenitor dataset, shown in the similar manner as in (**a**). Note that neither the CR, IR nor NR protomer was detected in these membrane openings by the TM analysis. Scale bars, 100 nm.

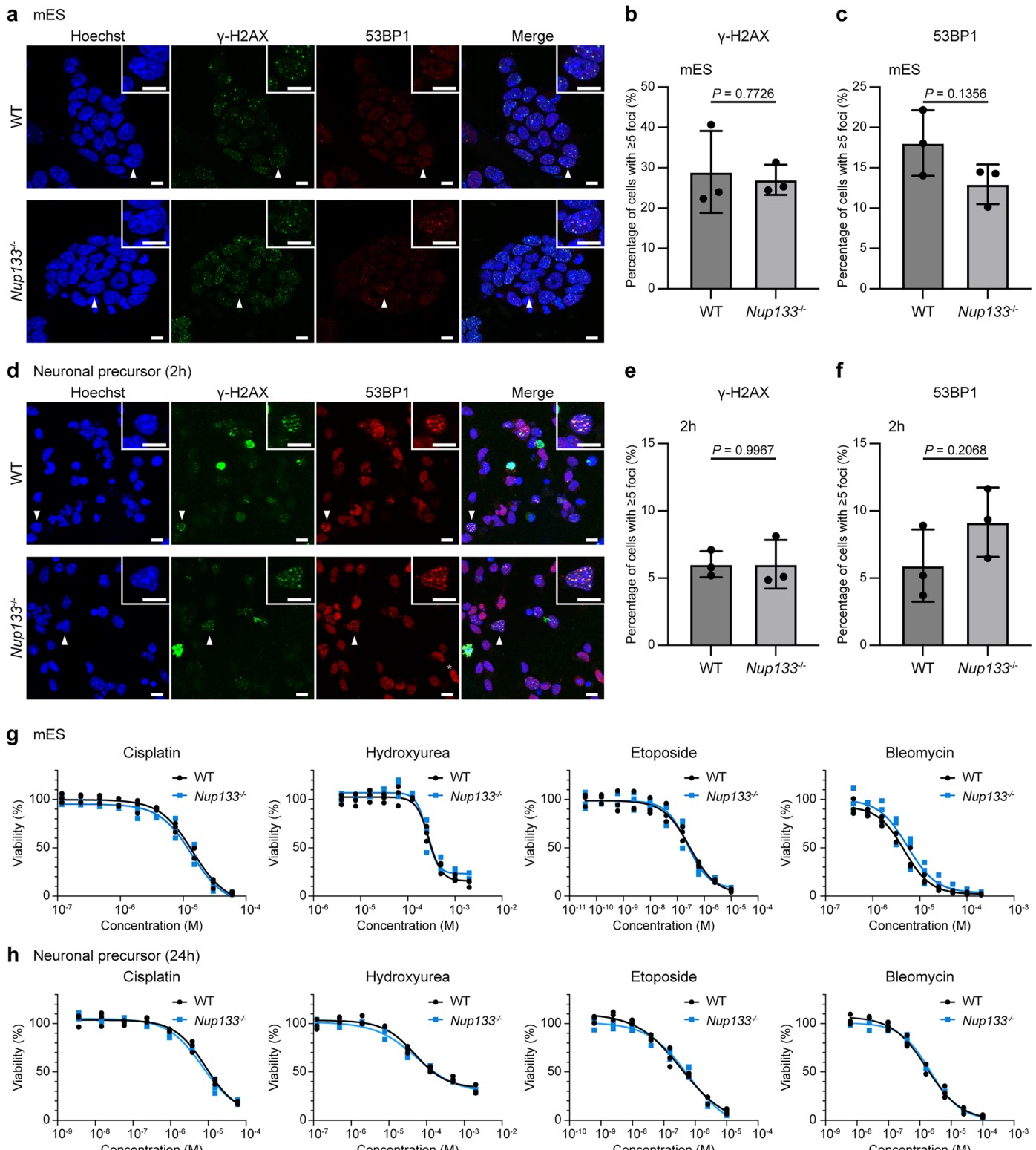

**Extended Data Fig. 8 | See next page for caption.**

**Extended Data Fig. 8 | Analysis of DNA damage in the mES cells and neuronal precursors.** (**a**) Representative immunofluorescence staining images of mES cells with γ-H2AX and 53BP1 foci. Maximum projection images of five central confocal frames are shown. Samples were prepared at 24 h after plating onto feeder cells. Examples of nuclei with ≥ 5 detected foci are indicated with white arrowheads and highlighted in the insets. Scale bars, 10 μm. (**b**, **c**) Mean percentage of nuclei with ≥ 5 γ-H2AX (**b**) and 53BP1 (**c**) foci in the wild-type and *Nup133*^-/- mES cells. Data are from three biological replicates, and at least 350 nuclei are analyzed for each replicate. Each data point represents the percentage calculated from individual dataset. Error bars denote standard deviations. Two-tailed Student's t-test. (**d**) Representative immunofluorescence staining images of neural progenitor cells 2 h after neuronal differentiation induction (hereafter also referred to as neuronal precursors for clarity) with γ-H2AX and 53BP1 foci.

Maximum projection images are shown. Examples of nuclei with ≥ 5 detected foci are indicated with white arrowheads and highlighted in the insets. Scale bars, 10 μm. (**e**, **f**) Mean percentage of nuclei with ≥ 5 γ-H2AX (**e**) and 53BP1 (**f**) foci in the wild-type and *Nup133*^-/- neuronal precursors 2 h after neuronal differentiation induction. Data are from three biological replicates, and at least 440 nuclei are analyzed for each replicate. Each data point represents the percentage calculated from individual dataset. Error bars denote standard deviations. Two-tailed Student's t-test. (**g**, **h**) Dose-response viability curves showing the viability percentage of mES cells (**g**) and neuronal precursors (**h**) treated with different concentrations of DNA-damaging agents. The dots show the measured values from three biological replicates. In (**h**), neuronal precursors were treated with DNA-damaging agents between 6 h and 24 h time points after neuronal differentiation induction. Source numerical data are available in source data.

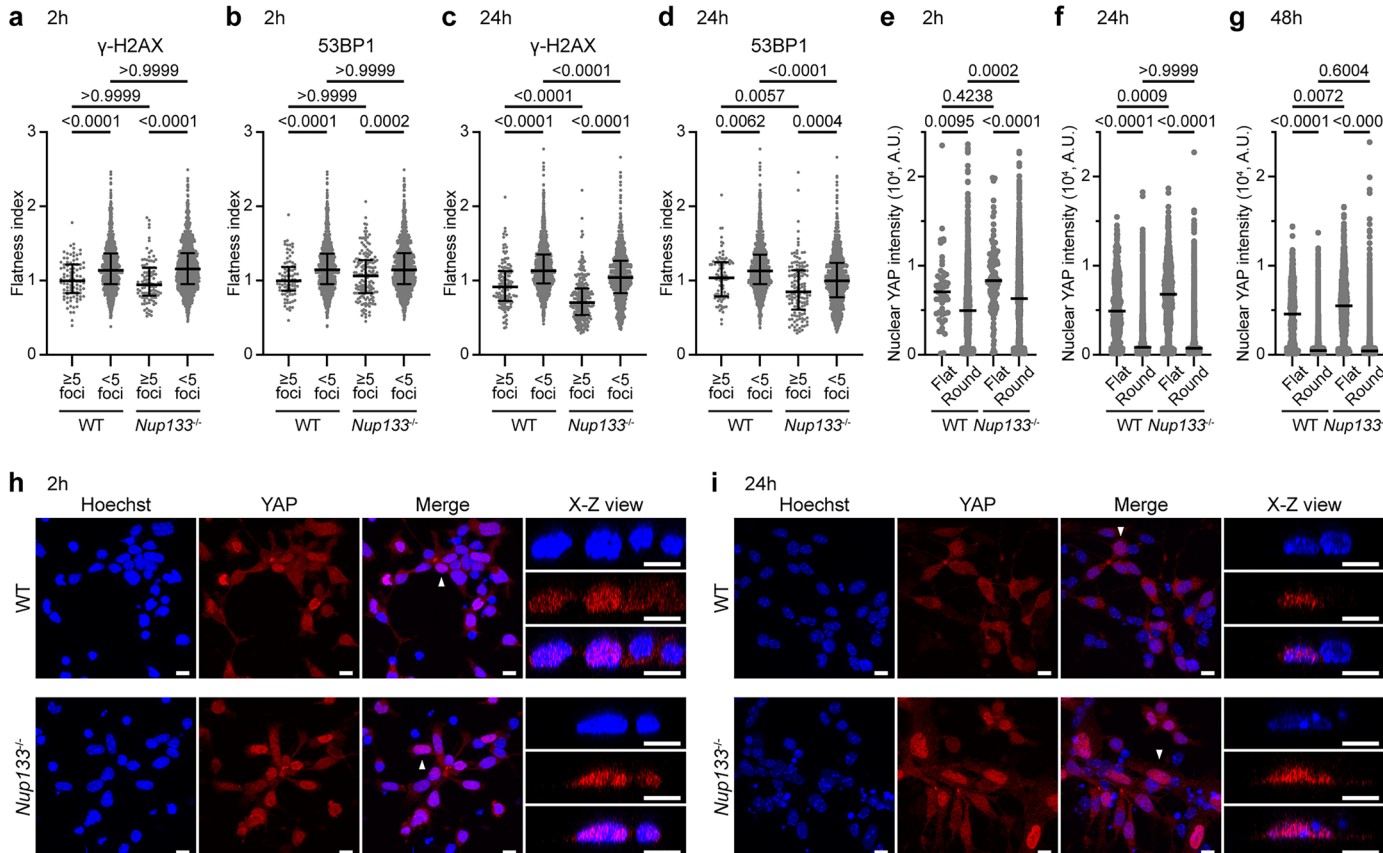

**Extended Data Fig. 9 | Analysis of nuclear morphology and nuclear YAP localization in the neuronal precursors. (a - d)** Quantification of nuclear flatness of neuronal precursors 2 h (**a, b**) and 24 h (**c, d**) after neuronal differentiation induction. The dataset is separated based on the presence of ≥ 5 γ-H2AX (**a, c**) and 53BP1 (**b, d**) foci. Flatness is calculated from the size of the bounding box of the segmented nuclei, and the ratio of its height and shorter horizontal axis is used as flatness index. Data from three biological replicates were combined. Each data point represents the flatness index calculated for individual nuclei. Bars denote the median values and the interquartile ranges. In (**a**), n = 91, 1418, 101, 1574 nuclei (from left to right) were analyzed. In (**b**), n = 87, 1422, 153, 1522 nuclei (from left to right) were analyzed. In (**c**), n = 139, 1821, 249, 1268 nuclei (from left to right) were analyzed. In (**d**), n = 90, 1870, 132, 1385 nuclei (from left to right) were analyzed. Kruskal-Wallis rank sum test followed by Dunn's multiple comparisons test. *P*-values are indicated above the plots. (**e - g**) Comparison of the mean nuclear YAP intensity in neuronal precursors at 2 h (**e**), 24 h (**f**), and 48 h (**g**) time

points after neuronal differentiation induction. Flatness index 0.8 is used as an arbitrary threshold, and nuclei with flatness index < 0.8 (flat) and ≥ 0.8 (round) are separately plotted. Data from two (**e**) or four (**f, g**) biological replicates were combined. Each data point represents the mean nuclear YAP intensity calculated for individual nuclei. Bars denote the median values. In (**e**), n = 49, 792, 101, 922 nuclei (from left to right) were analyzed. In (**f**), n = 537, 2100, 914, 1417 nuclei (from left to right) were analyzed. In (**g**), n = 448, 2474, 824, 1560 nuclei (from left to right) were analyzed. Kruskal-Wallis rank sum test followed by Dunn's multiple comparisons test. *P*-values are indicated above the plots. (**h, i**) Representative immunofluorescence staining images of YAP in neuronal precursors at 2 h (**h**) and 24 h (**i**) time points after neuronal differentiation induction. Maximum projection images are shown together with the X-Z orthogonal views of the nuclei highlighted with white arrowheads. Scale bars, 10 μm. Source numerical data are available in source data.

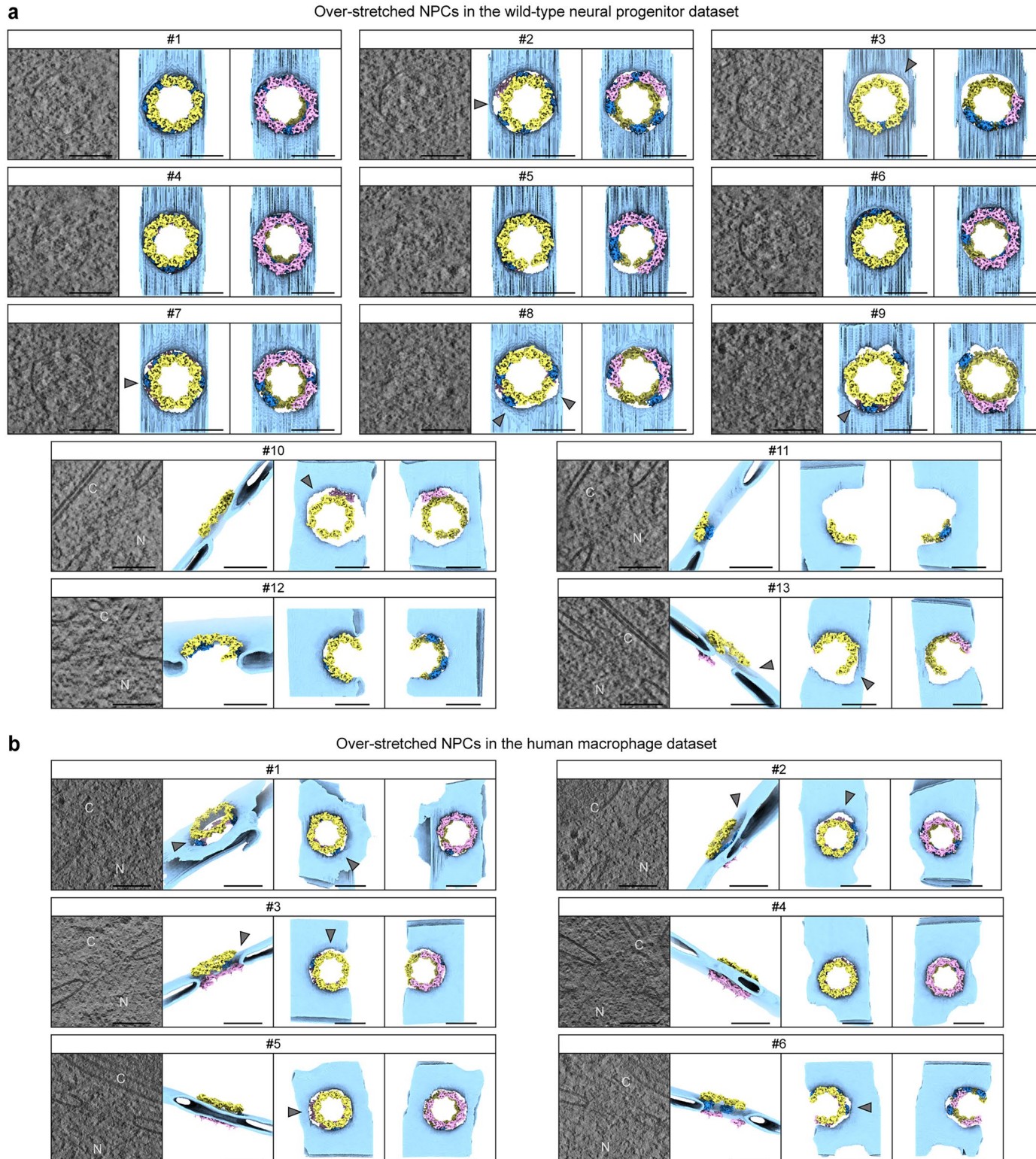

**Extended Data Fig. 10 | Over-stretched NPCs in the wild-type neural progenitor cells and human macrophages.** (**a**) Summary of the over-stretched NPCs observed in the wild-type neural progenitor dataset. Each example of an over-stretched NPC (#1–13) includes a representative slice through the tomograms (left), segmented membranes seen from the same orientation as in the left panel (second from the left), and segmented membranes seen from the cytoplasmic side (second from the right) and the nuclear side (right). The CR, IR and NR maps are projected back onto the segmented membranes, based on

the results of the TM analysis and colored as in Extended Data Fig. 7a. The CRs clearly detached from the outer nuclear membrane are highlighted with grey arrowheads. For the over-stretched NPCs oriented as top views in the tomograms (#1–9), the cytoplasmic views are omitted. C and N in the tomographic slices indicate the cytoplasm and nucleus, respectively. The example shown in Fig. 8a corresponds to #10. (**b**) Summary of the over-stretched NPCs observed in the human macrophage dataset, shown in the similar manner as in (**a**). The example shown in Fig. 8b corresponds to #2. Scale bars, 100 nm.

# Reporting Summary

## Statistics

For all statistical analyses, confirm that the following items are present in the figure legend, table legend, main text, or Methods section.

| n/a | Confirmed | |
|---|---|---|
| ☐ | ☒ | The exact sample size (*n*) for each experimental group/condition, given as a discrete number and unit of measurement |
| ☐ | ☒ | A statement on whether measurements were taken from distinct samples or whether the same sample was measured repeatedly |
| ☐ | ☒ | The statistical test(s) used AND whether they are one- or two-sided *Only common tests should be described solely by name; describe more complex techniques in the Methods section.* |
| ☒ | ☐ | A description of all covariates tested |
| ☒ | ☐ | A description of any assumptions or corrections, such as tests of normality and adjustment for multiple comparisons |
| ☐ | ☒ | A full description of the statistical parameters including central tendency (e.g. means) or other basic estimates (e.g. regression coefficient) AND variation (e.g. standard deviation) or associated estimates of uncertainty (e.g. confidence intervals) |
| ☐ | ☒ | For null hypothesis testing, the test statistic (e.g. *F*, *t*, *r*) with confidence intervals, effect sizes, degrees of freedom and *P* value noted *Give P values as exact values whenever suitable.* |
| ☒ | ☐ | For Bayesian analysis, information on the choice of priors and Markov chain Monte Carlo settings |
| ☒ | ☐ | For hierarchical and complex designs, identification of the appropriate level for tests and full reporting of outcomes |
| ☒ | ☐ | Estimates of effect sizes (e.g. Cohen's *d*, Pearson's *r*), indicating how they were calculated |

*Our web collection on statistics for biologists contains articles on many of the points above.*

## Software and code

Policy information about availability of computer code

| Data collection | SerialEM(3.8.5) |
|---|---|
| Data analysis | gctf(1.06), IMOD(version 4.10.9 and 4.11.5), AreTomo (1.33), novaCTF (https://github.com/turonova/novaCTF), novaSTA (https://github.com/turonova/novaSTA), Dynamo(1.1.532), Relion(version 3.1), STOPGAP(version 0.7.1, https://github.com/wan-lab-vanderbilt/STOPGAP), MATLAB R2019b/R2022b, ChimeraX (1.5),  Membrain-seg (https://github.com/teamtomo/membrain-seg), Amira Software (Thermo Scientific, ver 2023.2), ArtiaX, napari, FiJi (ImagJ version 1.54f), GraphPad Prism 9 |

For manuscripts utilizing custom algorithms or software that are central to the research but not yet described in published literature, software must be made available to editors and reviewers. We strongly encourage code deposition in a community repository (e.g. GitHub). See the Nature Portfolio guidelines for submitting code & software for further information.

## Data

Policy information about availability of data

All manuscripts must include a data availability statement. This statement should provide the following information, where applicable:

- Accession codes, unique identifiers, or web links for publicly available datasets
- A description of any restrictions on data availability
- For clinical datasets or third party data, please ensure that the statement adheres to our policy

Our final data availability statement says "The subtomogram averages described in this study are deposited in the Electron Microscopy Data Bank (EMDB) with the

# Research involving human participants, their data, or biological material

Policy information about studies with human participants or human data. See also policy information about sex, gender (identity/presentation), and sexual orientation and race, ethnicity and racism.

| | |
|---|---|
| Reporting on sex and gender | This study does not involve human research participants. The cryo-ET dataset on primary human macrophages we analyzed was acquired as part of another study (doi:10.1016/j.cell.2024.12.008), where the sample preparation is described in detail. In brief, Monocyte-derived macrophages (MDMs) were obtained from human peripheral blood mononuclear cells (PBMC) isolated from buffy coats of anonymous blood donors at the Heidelberg University Hospital Blood Bank according to the regulations of the local ethics committee. |
| Reporting on race, ethnicity, or other socially relevant groupings | not applicable, see above |
| Population characteristics | not applicable, see above |
| Recruitment | not applicable, see above |
| Ethics oversight | Local ethics committee at the Heidelberg University Hospital |

Note that full information on the approval of the study protocol must also be provided in the manuscript.

# Field-specific reporting

Please select the one below that is the best fit for your research. If you are not sure, read the appropriate sections before making your selection.

☒ Life sciences          ☐ Behavioural & social sciences          ☐ Ecological, evolutionary & environmental sciences

For a reference copy of the document with all sections, see nature.com/documents/nr-reporting-summary-flat.pdf

# Life sciences study design

All studies must disclose on these points even when the disclosure is negative.

| | |
|---|---|
| Sample size | We did not pre-determine the sample size when the study was being designed. We prepared three to six different grids for each sample and acquired the maximum amount of cryo-ET data within the available machine time. For the other experiments, we determined the sample size based on similar experiments performed in previous literatures and the available time for data acquisition. |
| Data exclusions | None. |
| Replication | The Cryo-ET data of the wild-type and Nup133KO mES cells were acquired from six EM grids prepared in three and two independent plunge freezing sessions, respectively. The cryo-ET data of the neural progenitor samples were acquired from three EM grids prepared in three independent plunge freezing sessions. All the data from different data acquisition sessions show consistent results. Differentiation experiments were repeated seven times and consistent results were obtained. Immunofluorescence staining experiments were performed at least three times, and all the attempts were successful. |
| Randomization | We did not use randomization in our study. For the experiments involving mES cells, the wild-type and Nup133KO cell lines were always compared under the same conditions throughout the study, and all the experiments were conducted for both cell lines using the same type of dishes and same batch of reagents to minimize experimental biases. The data analyses were also conducted in the same way between two cell lines. |
| Blinding | We did not use blinding in our study. For both the EM and light microscopy imaging experiments, blinding was not practical, because Nup133KO-specific phenotypes were in many cases readily visible in the datasets. For structural analyses, blinding is not applicable due to the necessity of combining the data from the same cell line. |

# Reporting for specific materials, systems and methods

We require information from authors about some types of materials, experimental systems and methods used in many studies. Here, indicate whether each material, system or method listed is relevant to your study. If you are not sure if a list item applies to your research, read the appropriate section before selecting a response.

## Materials & experimental systems

| n/a | Involved in the study |
|---|---|
| ☐ | ☒ Antibodies |
| ☐ | ☒ Eukaryotic cell lines |
| ☒ | ☐ Palaeontology and archaeology |
| ☒ | ☐ Animals and other organisms |
| ☒ | ☐ Clinical data |
| ☒ | ☐ Dual use research of concern |
| ☒ | ☐ Plants |

## Methods

| n/a | Involved in the study |
|---|---|
| ☒ | ☐ ChIP-seq |
| ☒ | ☐ Flow cytometry |
| ☒ | ☐ MRI-based neuroimaging |

## Antibodies

| | |
|---|---|
| Antibodies used | mouse anti-Pax6 (Developmental Studies Hybridoma Bank, RRID AB_528427, dilution 1:12), mouse anti-Oct3/4 (Santa Cruz Biotechnology, sc-5279, dilution 1:200), mouse anti-γ-H2AX (Merck, 05-636, dilution 1:1000), rabbit anti-53BP1 (Novus Biologicals, NB100-304SS, dilution 1:1000), rabbit anti-YAP (Cell Signaling Technology, #14074, dilution 1:100), goat anti-mouse Alexa Fluor Plus 488-conjugated antibody (Thermo Fisher Scientific, A32723, dilution 1:2000), goat anti-rabbit Alexa Fluor 594-conjugated antibody (Thermo Fisher Scientific, A11012, dilution 1:2000) |
| Validation | All the antibodies were directly purchased from the supplier. All the primary antibodies have been used multiple times in previous literatures, and were validated for species reactivity and application by each supplier with following validation information;<br>Anti-Pax6 (DSHB, AB_528427):<br>Reacts to amphibian, avian, fish, human, lizard, mouse, opossum, planaria, rat, turtle, zebrafish; Suitable for ChIP, FACS, FFPE, Function Blocking, Gel Supershift, IF, IHC, IP, WB<br>Anti-Oct3/4 (Santa Cruz, sc-5279):<br>Reacts to mouse, rat and human; Suitable for WB, IP, IF, IHC(P), FCM and ELISA<br>Anti-γ-H2AX (Merck, 05-636):<br>Reacts to vertebrates; Suitable for ICC, IF, WB, ChIP, IHC<br>Anti-53BP1 (Novus Biologicals, NB100-304SS):<br>Reacts to human, mouse, rat, bovine, canine, fish, goat, primate, porcine, feline, rabbit, sheep; Suitable for WB, ChIP, Flow, Func, ICC/IF, IHC, IP, WB, ChIP<br>Anti-YAP (Cell Signaling Technology, #14074):<br>Reacts to human, mouse, rat, hamster, monkey; Suitable for WB, IP, IHC, IF, FCM, ChIP,  CUT & RUN |

## Eukaryotic cell lines

Policy information about cell lines and Sex and Gender in Research

| | |
|---|---|
| Cell line source(s) | Nup133-knock out mES cell line was obtained from HM1 mES cell line. It was originally established and validated in Valérie Doye group at the Institut Jacques Monod (Souquet et al., Cell Reports, 2018). |
| Authentication | Cell lines were regularly authenticated based on morphology. To minimize the risk of cross contamination and spontaneous differentiation of mES cells, we only maintained cells in culture for less than 10 passages. |
| Mycoplasma contamination | Cell lines were tested negative for mycoplasma contamination. |
| Commonly misidentified lines (See ICLAC register) | No commonly misidentified cell lines were used in this study. |

## Plants

| | |
|---|---|
| Seed stocks | This study does not involve plant samples. |
| Novel plant genotypes | This study does not involve plant samples. |
| Authentication | This study does not involve plant samples. |

