## [Peer Review File · Nature Cell Biology]

Nuclear pores safeguard the integrity of the nuclear envelope

Corresponding Author: Professor Martin Beck

Version 0:

Decision Letter:

*Please delete the link to your author homepage if you wish to forward this email to co-authors.

Dear Martin,

Thank you again for submitting your manuscript, "Nuclear pores safeguard the integrity of the nuclear envelope", to Nature Cell Biology. It has now been seen by 3 referees, who are experts in mechanobiology (Referee #1); NPC, age/disease (Referee #2); and cryoEM-ET (Referee #3). As you will see from their comments (attached below), they found the work of potential interest, but have raised substantial concerns that in our view would need to be addressed with considerable revisions before we can consider publication in Nature Cell Biology.

As per our standard editorial process, Nature Cell Biology editors discuss the referee reports in detail within the editorial team, including the chief editor, to identify key referee points that should be addressed with priority, as opposed to requests that are beyond the scope of the current study. To guide the scope of the revisions, I have listed these points below. Our standard revision period is six months and we are committed to providing a fair and constructive peer-review process, so please feel free to contact me if you would like to discuss any of the referee comments further or if you anticipate any issues or delays addressing the reviews.

In particular, it would be essential to address the following points:

1— The reviewers consistently called for more detailed analyses of the DNA damage phenotypes:

Rev#1 point #3
Rev#2 minor point #1
Rev#3 point #7

2- We agree with Reviewer #1 that additional studies of the contribution of mechanical stress and its impact on the NE and NPC are important to support the functional conclusions (points #1-2).

3- Lastly, please also address the reviewers' comments about strengthening or clarifying existing data, clarity and organization in presentation and analysis, requests for methodological details and discussion or text edits.

4- Finally, please pay close attention to our guidelines on statistical and methodological reporting (listed below) as failure to do so may delay the reconsideration of the revised manuscript. In particular, please provide:

We would be happy to consider a revised manuscript that would satisfactorily address these points, unless a similar paper is published elsewhere or is accepted for publication in Nature Cell Biology in the meantime.

- ensure that it conforms to our format instructions and publication policies (see below and <https://www.nature.com/nature/for->

authors).

- provide a point-by-point rebuttal to the full referee reports verbatim, as provided at the end of this letter.

- provide the completed Reporting Summary (found here <https://www.nature.com/documents/nr-reporting-summary.pdf>). This is essential for reconsideration of the manuscript will be available to editors and referees in the event of peer review. For more information see <http://www.nature.com/authors/policies/availability.html> or contact me.

Nature Cell Biology is committed to improving transparency in authorship. As part of our efforts in this direction, we are now requesting that all authors identified as 'corresponding author' on published papers create and link their Open Researcher and Contributor Identifier (ORCID) with their account on the Manuscript Tracking System (MTS), prior to acceptance. ORCID helps the scientific community achieve unambiguous attribution of all scholarly contributions. You can create and link your ORCID from the home page of the MTS by clicking on 'Modify my Springer Nature account'. For more information please visit www.springernature.com/orcid.

This journal strongly supports public availability of data. Please place the data used in your paper into a public data repository, or alternatively, present the data as Supplementary Information. If data can only be shared on request, please explain why in your Data Availability Statement, and also in the correspondence with your editor. Please note that for some data types, deposition in a public repository is mandatory - more information on our data deposition policies and available repositories appears below.

Link Redacted

We hope that you will find our referees' comments and editorial guidance helpful. Please do not hesitate to contact me if there is anything you would like to discuss. Thank you again for considering the journal for your work.

Best wishes,

Melina

Melina Casadio, PhD
Senior Editor, Nature Cell Biology
ORCID ID: <https://orcid.org/0000-0003-2389-2243>

Reviewers' Comments:

Reviewer #1:

Remarks to the Author:

The manuscript by Taniguchi and co-workers analyze the structural changes of the nuclear pore upon ES cell differentiation into neural progenitors. While they observe no drastic changes, they find compositional changes and in particular a role for Nup133 in determining nuclear pore structure and integrity in neuronal progenitors. Deletion of Nup133 has been previously reported to lead to neuronal differentiation defects and the authors now report an important role in nuclear pore complex architecture and stability. Overall the manuscript describes intriguing and potentially important findings that will be interesting to the wider cell biology community. The manuscript is well written and the data largely of high quality.

A few issues should be addressed prior to publication, mainly related to the correlations between nuclear envelope tension,

nuclear pore site fragility and DNA damage that appears somewhat underdeveloped and mostly correlative at this point.

1. The authors point to increased mechanical tension as the cause for nuclear pore fragility. However, it is not clear why this should be cell type specific. While the authors show pore dilation, this could be a result of tension-unrelated factors. Thus, the authors should address if NE tension or forces acting on the NE are in fact higher in neurons than in ES cells. Comparative quantitative analyses of NE tension/mechanics would greatly strengthen the manuscript.

2. The authors propose that increase in mechanical stress triggers the nuclear pore rupture. This should be demonstrated by triggering nuclear deformation by for example hypotonic shock

3. It is not clear why this nuclear pore fragility would cause widespread DNA damage. Is the DNA herniating into the cytoplasm? Is there micronucleus formation? Or could the effect be more indirect through regulation of transcription/replication conflicts for example that are known to be modulated through nuclear pore-related processes. The authors should analyse cGAS activation to assess DNA herniations as well as more carefully analyse nuclear shapes and micronuclear formation as well as address the role of replication stress, in particular as the analyses are done in S-phase synchronized cells.

Reviewer #2:

Remarks to the Author:

The manuscript by Taniguchi et al. examined NPC architecture in mouse embryonic stem cells (mES) and in neural progenitor cells in both wild-type and Nup133-deficient cells, using state-of-the-art cryo-electron tomography-based structural biology. Previous studies had shown that ESCs are unaffected without Nup133, while differentiated neurons are impacted. Additionally, prior research demonstrated that differentiation correlates with increased mechanical stress, and that NPCs can dilate upon mechanical stress. The current study provides a structural explanation for these observations by investigating the relationship between NPC diameter changes, cell differentiation, and disruption of NPC scaffold architecture. The hypothesis being that the NPC scaffold adapts to mechanical changes in the nuclear envelope during differentiation, with impaired responses in perturbed NPC architectures.

The study presents intriguing insights into NPCs lacking Nup133! I find it very exciting to see that in Nup133 ^{-/-} cells some 5% of NPCs have an altered rotational symmetry and, moreover, that a large fraction of NPCs have incomplete nuclear and cytoplasmic rings! These conclusions are drawn from three-dimensional template matching (TM). The incomplete presence of the CR and NR is proposed to weaken the ability of the NPC to adapt to forces stretching the nuclear envelope. Their incomplete presence correlates with the appearance of overstretched NPCs, with some NPCs losing connection to the pore membrane. These openings compromise nuclear compartmentalization, potentially leading to increased DNA damage. Overall, this is a very exciting and important structural study with – as far as I can judge – high quality data. The findings are important for the NPC community and researchers in developmental biology, neurobiology, and possibly cancer biology fields. The relationship to NE ruptures happening at NPCs is an exciting discussion point.

I enthusiastically support publishing this paper and only have a few minor suggestions for improving the manuscript.

Most importantly I feel that the phrasing in the text suggests at several places that the authors have data to support that the NPCs were fine at first, but that the NR and CRs disintegrate and detach as a result of mechanical stress (over-stretching) during differentiation. E.g., in line 316 they write “we conclude that an increase in nuclear envelope tension leads to NPC over-stretching, which can result in disintegration of NPCs” While that is not un-logical it is also not proven and it would be better to make that clear. If the NPCs disintegrate or even misassemble independent of stretching, all of their observations would still stand, e.g., their diameter would also increase.

This uncertainty also makes me somewhat less enthusiastic about the title which puts safeguarding nuclear integrity at the central place, while the data is most directly on NPC structural integrity.

Other minor comments:

- 1) The connection to DNA damage could be elucidated further, particularly regarding the link between decreased nuclear compartmentalization due to unstable Nup113- Δ NPCs and increased γ -H2AX foci. Do the authors think there are more γ -H2AX foci because there is more damage or because the repair is impaired? How would either relate to NE ruptures or a compromised NPC permeability barrier? This is not so clear from the existing literature. Including the publications cited.
- 2) Figure 1A,B, S3, Figure 2: it would be useful to point out the protrusion from the NR at the Nup107/Nup133 region, which the authors write “likely corresponding to nuclear basket filaments.” It is exciting in itself that these filaments are resolved and also that this protrusion is less clear in the nup133 ^{-/-}.
- 3) Regarding the detachment of the CRs reported in (Fig. 6B, S6A), some images may not clearly show this detachment, such as #13 and #15.
- 4) P 7: Consider rephrasing “Despite this critical scaffolding role and its clear presence within the averages of the wild-type NPC (Fig.136 1B), Nup133 is dispensable for the proliferation of mES cells”, removing the technical interruption regarding its clear presence within wild-type NPC averages.
- 5) Does the new structural data provide an explanation why Nup133 deletion results in clustering of NPCs in yeast?

Reviewer #3:

Remarks to the Author:

Nuclear pore complexes (NPCs) regulate exchange of molecules between the nuclear and cytoplasmic environments. NPCs are also an integral part of the overall architecture of the nucleus, which houses the genetic contents of cells. Mutations in NPC subunit proteins (nucleoporins or Nups) contribute to many diseases, but the mechanisms of action are unclear. Taniguchi and colleagues investigate the deletion of Nup133 in mouse embryonic stem cells and use this as a system to investigate how Nup133 deletion alters NPC architecture during differentiation. I find this study to be fantastic, and a foundational building block for the field to understand how disease mutations alter cellular differentiation via introducing a plethora of architectural variability within NPCs. Moreover, the authors use their observations to generate key hypotheses on how fundamental the architectural plasticity of NPCs is for development. Furthermore, the structural work performed here is world class. This paper is absolutely fit for Nature Cell Biology. I congratulate the authors on these findings, which are broadly interesting to those in the NPC field, cellular development, disease pathologies, and cryo-ET. I include comments below.

1. Fig. S1D - an example raw image of cells stained with the Pax6 antibody should be shown prior to and after differentiation for WT and Nup133 deletion.
2. Fig 1C - is it possible for the authors to measure the curvature change between mES and differentiated progenitor?
3. Fig 1B - the Nup133 region of the map would be more clear if the authors colored the zone of the map.
4. Page 9, line 185 - should Figure 2 here actually refer to Figure 3?
5. Figure 3 - This figure would be strengthened by plotting the overall populations of each symmetry group here to highlight the differences between WT and Nup133 del.
6. In general, I find the high resolution template matching to be quite powerful in showing the degree of heterogeneity in the Nup133 del in each layer of the NPCs.
7. Fig 7 - the authors should compare the change in H2AX foci between mES and differentiated progenitor to strengthen the point. It is important to know the extent of DNA damage prior to differentiation in this context.
8. It would be interesting for the authors to speculate more in the discussion on the functional consequences of the symmetry mismatches they observe via TM in CRs and NRs. Is the functional consequence of this only envelope integrity? How would this affect transport across the NPC?

Methods should be written concisely, but should contain all elements necessary to allow interpretation and replication of the results. As a guideline, Methods sections typically do not exceed 3,000 words. The Methods should be divided into subsections listing reagents and techniques. When citing previous methods, accurate references should be provided and any alterations should be noted. Information must be provided about: antibody dilutions, company names, catalogue numbers and clone numbers for monoclonal antibodies; sequences of RNAi and cDNA probes/primers or company names and catalogue numbers if reagents are commercial; cell line names, sources and information on cell line identity and authentication. Animal studies and experiments involving human subjects must be reported in detail, identifying the committees approving the protocols. For studies involving human subjects/samples, a statement must be included confirming that informed consent was obtained. Statistical analyses and information on the reproducibility of experimental results should be provided in a section titled "Statistics and Reproducibility".

All Nature Cell Biology manuscripts submitted on or after March 21 2016 must include a Data availability statement as a separate section after Methods but before references, under the heading "Data Availability". For Springer Nature policies on data availability see <http://www.nature.com/authors/policies/availability.html>; for more information on this particular policy see <http://www.nature.com/authors/policies/data/data-availability-statements-data-citations.pdf>. The Data availability statement should include:

- Accession codes for primary datasets (generated during the study under consideration and designated as "primary accessions") and secondary datasets (published datasets reanalysed during the study under consideration, designated as "referenced accessions"). For primary accessions data should be made public to coincide with publication of the manuscript. A list of data types for which submission to community-endorsed public repositories is mandated (including sequence, structure, microarray, deep sequencing data) can be found here <http://www.nature.com/authors/policies/availability.html#data>.
- Unique identifiers (accession codes, DOIs or other unique persistent identifier) and hyperlinks for datasets deposited in an approved repository, but for which data deposition is not mandated (see here for details <http://www.nature.com/sdata/data-policies/repositories>).
- At a minimum, please include a statement confirming that all relevant data are available from the authors, and/or are included with the manuscript (e.g. as source data or supplementary information), listing which data are included (e.g. by figure panels and data types) and mentioning any restrictions on availability.
- If a dataset has a Digital Object Identifier (DOI) as its unique identifier, we strongly encourage including this in the Reference list and citing the dataset in the Methods.

We recommend that you upload the step-by-step protocols used in this manuscript to the Protocol Exchange. More details can found at www.nature.com/protocolexchange/about.

All imaging data should be accompanied by scale bars, which should be defined in the legend.

Cropped images of gels/blots are acceptable, but need to be accompanied by size markers, and to retain visible background signal within the linear range (i.e. should not be saturated). The boundaries of panels with low background have to be demarked with black lines. Splicing of panels should only be considered if unavoidable, and must be clearly marked on the figure, and noted in the legend with a statement on whether the samples were obtained and processed simultaneously. Quantitative comparisons between samples on different gels/blots are discouraged; if this is unavoidable, it should only be performed for samples derived from the same experiment with gels/blots were processed in parallel, which needs to be stated in the legend.

Unprocessed scans of all key data generated through electrophoretic separation techniques need to be presented in a supplementary figure that should be labelled and numbered as the final supplementary figure, and should be mentioned in every relevant figure legend. This figure does not count towards the total number of figures and is the only figure that can be displayed over multiple pages, but should be provided as a single file, in PDF or TIFF format. Data in this figure can be

displayed in a relatively informal style, but size markers and the figures panels corresponding to the presented data must be indicated.

The total number of Supplementary Figures (not including the “unprocessed scans” Supplementary Figure) should not exceed the number of main display items (figures and/or tables (see our Guide to Authors and March 2012 editorial <http://www.nature.com/ncb/authors/submit/index.html#suppinfo>; <http://www.nature.com/ncb/journal/v14/n3/index.html#ed>). No restrictions apply to Supplementary Tables or Videos, but we advise authors to be selective in including supplemental data.

GUIDELINES FOR EXPERIMENTAL AND STATISTICAL REPORTING

REPORTING REQUIREMENTS – We are trying to improve the quality of methods and statistics reporting in our papers. To that end, we are now asking authors to complete a reporting summary that collects information on experimental design and reagents. The Reporting Summary can be found here <https://www.nature.com/documents/nr-reporting-summary.pdf>. If you would like to reference the guidance text as you complete the template, please access these flattened versions at <http://www.nature.com/authors/policies/availability.html>.

We strongly recommend the presentation of source data for graphical and statistical analyses as a separate Supplementary Table, and request that source data for all independent repeats are provided when representative experiments of multiple independent repeats, or averages of two independent experiments are presented. This supplementary table should be in Excel format, with data for different figures provided as different sheets within a single Excel file. It should be labelled and numbered as one of the supplementary tables, titled “Statistics Source Data”, and mentioned in all relevant figure legends.

Version 1:

Decision Letter:

Our ref: NCB-A53593A

6th December 2024

Dear Dr. Beck,

Thank you for submitting your revised manuscript “Nuclear pores safeguard the integrity of the nuclear envelope” (NCB-A53593A). It has now been seen by the original referees and their comments are below. The reviewers find that the paper has improved in revision, and therefore we'll be happy in principle to publish it in Nature Cell Biology, pending minor revisions to satisfy the referees' final requests and to comply with our editorial and formatting guidelines.

We are now performing detailed checks on your paper and will send you a checklist detailing our editorial and formatting requirements in about 1-2 weeks. Please do not upload the final materials and make any revisions until you receive this additional information from us.

Thank you again for your interest in Nature Cell Biology. Please do not hesitate to contact me if you have any questions.

Sincerely,

Melina

Melina Casadio, PhD
Senior Editor, Nature Cell Biology
Consulting Editor, Nature Structural & Molecular Biology
ORCID ID: <https://orcid.org/0000-0003-2389-2243>

Reviewer #1 (Remarks to the Author):

The authors have made substantial efforts to address concerns of this reviewer. While some aspects (source of DNA damage, nuclear envelope tension) remain partially unanswered, these are challenging questions that require additional in-depth studies and I appreciate that authors transparency in expressing limitations of their approaches. The manuscript is in good shape for publication.

Reviewer #2 (Remarks to the Author):

The additional cell biological data on 53BP1 foci, CGAs, BAF and YAP localization, flatness and sensitivity to DNA-damaging agents further strengthened the conclusions of the work. The textual edits take care of resolving some unclaritys. I enthusiastically support publication of the work.

a minor suggestion would be to indicate in figure 7 and extended data 9 which cells are considered to have less than 5 foci, as just judging by eye, the images may not be representative of the quantification.

Reviewer #3 (Remarks to the Author):

The authors addressed my concerns, I support publication.

Version 2:

Decision Letter:

Dear Dr. Beck,

Thank you for your patience. I am pleased to inform you that your manuscript, "Nuclear pores safeguard the integrity of the nuclear envelope", has now been accepted for publication in Nature Cell Biology.

Please note that *Nature Cell Biology* is a Transformative Journal (TJ). Authors may publish their research with us through the traditional subscription access route or make their paper immediately open access through payment of an article-processing charge (APC). Authors will not be required to make a final decision about access to their article until it has been accepted. [Find out more about Transformative Journals](https://www.springernature.com/gp/open-research/transformative-journals)

Authors may need to take specific actions to achieve [compliance with funder and institutional open access mandates](https://www.springernature.com/gp/open-research/funding/policy-compliance-faqs). If your research is supported by a funder that requires immediate open access (e.g. according to [Plan S principles](https://www.springernature.com/gp/open-research/plan-s-compliance)) then you should select the gold OA route, and we will direct you to the compliant route where possible. For authors selecting the subscription publication route, the journal's standard licensing terms will need to be accepted, including [self-archiving policies](https://www.springernature.com/gp/open-research/policies/journal-policies). Those licensing terms will supersede any other terms that the author or any third party may assert apply to any version of the manuscript.

If you have not already done so, we strongly recommend that you upload the step-by-step protocols used in this manuscript to protocols.io (<https://protocols.io>), an open online resource that allows researchers to share their detailed experimental know-how. All uploaded protocols are made freely available and are assigned DOIs for ease of citation. Protocols and Nature Portfolio journal papers in which they are used can be linked to one another, and this link is clearly and prominently visible in the online versions of both. Authors who performed the specific experiments can act as primary authors for the Protocol as they will be best placed to share the methodology details, but the Corresponding Author of the present research paper should be included as one of the authors. By uploading your Protocols onto protocols.io, you are enabling researchers to more readily reproduce or adapt the methodology you use, as well as increasing the visibility of your protocols and papers. You can also establish a dedicated workspace to collect your lab Protocols. Further information can be found at <https://www.protocols.io/help/publish-articles>.

Nature Cell Biology encourages authors presenting evidence for cell, biological, molecular, and genetic interactions to consider communicating these findings using Biofactoid (<https://biofactoid.org/>). This tool helps users share a searchable representation of interactions (e.g. binding, gene expression, post-translational modification) between genes, gene products, or chemicals. Information added to Biofactoid, with author attribution, is shared on social media and public databases, such as Pathway Commons, where it can be discovered and analyzed in the context of a large and growing corpus of knowledge.

Best regards,

George Inglis

George Inglis, PhD
Senior Editor

[Research Cross-Journal Editorial Team](https://www.nature.com/ncb/research-cross-journal-editorial-team)
Nature Cell Biology

** Visit the Springer Nature Editorial and Publishing website at http://editorial-jobs.springernature.com?utm_source=ejp_NCB_email&utm_medium=ejp_NCB_email&utm_campaign=ejp_NCB for more information about our career opportunities. If you have any questions please click [here](mailto:editorial.publishing.jobs@springernature.com).

Point-by-point response to the reviewers' comments

We thank all reviewers and the editor for their constructive comments, which helped us to further improve the manuscript.

General response:

1. The reviewers and the editor consistently asked for a more detailed analyses of the DNA damage phenotype.

To address these concerns regarding the increase in γ -H2AX foci that we observe in *Nup133*^{-/-} cells, we carried out the additional experiments described below.

- We included additional analyses, and now show that 53BP1 foci also accumulate in *Nup133*^{-/-} neural progenitor cells undergoing neuronal differentiation (for clarity, these differentiating neural progenitor cells are referred to as “neuronal precursors” throughout the text), together with γ -H2AX foci. These data support the conclusion that DNA damage and repair more often occurs in *Nup133*^{-/-} cells when they undergo neuronal differentiation (Fig. 7a-c).
- We furthermore show that there is a subpopulation of neuronal precursors with an unusually flat and stretched out nuclear morphology (Fig. 7d) and cells with nuclear YAP localization (Fig. 7e). The abundance of such cells is increased in *Nup133*^{-/-} neuronal precursors compared to wildtype at both 24 h and 48 h time points after differentiation induction (Fig. 7f-k, Extended Data Fig. 9). These data support the idea that the nuclear envelope of differentiating neuronal precursors is exposed to a higher mechanical load and that the *Nup133*^{-/-} neuronal precursors are more severely affected by such stress.
- To exclude that *Nup133*^{-/-} mES cells have a general DNA damage repair defect, we compared their sensitivity to different types of DNA damaging agents (cisplatin, hydroxyurea, etoposide, bleomycin) with that of wildtype mES cells. We found that there was no difference (Extended Data Fig. 8g), indicating that canonical DNA repair pathways are likely not affected by Nup133 deletion. We also did not observe a clear difference for the neuronal precursors undergoing neuronal differentiation (Extended Data Fig. 8h), arguing that the differentiation protocol used here is unlikely to grossly alter DNA repair capability of *Nup133*^{-/-} cells.
- As reviewer #3 suggested, we also determined the number of γ -H2AX and 53BP1 foci in unperturbed mES cells (Extended Data Fig. 8a-c). We did not find a striking difference between wildtype and *Nup133*^{-/-} cells, indicating that the increased accumulation of DNA damage in the *Nup133*^{-/-} cells in comparison to the wildtype cells is observed specifically during neuronal differentiation.
- We also analyzed cGAS and BAF localization in wildtype and *Nup133*^{-/-} neuronal precursors. We did not observe a marked accumulation of any marker at the nuclear envelope, as would have been expected in the case of larger DNA spillage from the nucleus into the cytoplasm. Based on visual inspection of the light microscopy images, there were also no indications for the presence of micronuclei in our dataset.

Taken together, we interpret this data in a way that the NPC disintegration events that we observe in the tomograms of the *Nup133*^{-/-} neural progenitor cells are likely early stages of nuclear integrity loss and subsequent nuclear envelope rupture. With those disintegrated

NPCs, no large openings nor a drastic loss of the nucleus-cytoplasmic barrier function would be present, which is consistent with the fact that we do not observe massive morphological changes directly upon induction of neuronal differentiation. Also in line with this interpretation is the fact that we reproducibly detect a significant amount of DNA damage by analysis of γ -H2AX and 53BP1 foci in differentiating neuronal precursors. We have carefully revised the text to make sure this becomes clear to the reader.

2. Reviewer #1 and the editor requested further investigation of the contribution of mechanical stress and its impact on the NE and NPC.

We can reconcile the reviewer's concern and have performed several experiments, which however did not yield conclusive results. As outlined in the response to the first point, our current interpretation of the data is that NPC disintegration occurs at an early stage of nuclear integrity loss. Thus, no major effects on nuclear-cytoplasmic barrier integrity can be seen. This prevents us from using classical cell biological assays such as GFP-NLS constructs or the markers discussed above, as readouts.

We nonetheless attempted a confinement experiment, where wildtype and *Nup133*^{-/-} mES cells expressing 3xGFP-NLS or mCherry-BAF were seeded without feeder cells as single cell culture (with LIF to promote stem cell self-renewal) and subjected to mechanical pressure using a 4D-cell mechanical pressure system while being monitored in live-cell imaging. However, we did not observe conclusive or consistent effects. During the course of these experiments, we realized that the stem cell population behaves very heterogeneously when seeded as a monolayer without feeder cells (both in wildtype and *Nup133*^{-/-}). This heterogeneity includes differences in terms of nuclear size, shape, degree of clustering, and adhesion to the cell culture surface. This is likely due to the inherent properties of mES cells to form clusters and to spontaneously differentiate when cultured without feeder cells. It is conceivable that such heterogeneous cells simply respond very differentially to mechanical challenges. In addition, we attempted the confinement experiment with neural progenitor samples. However, these neural progenitor cells are incompatible with transfection, and the marker molecules could not be sufficiently expressed. We thus conclude that the experimental setup for the confinement assay is incompatible with our samples.

Instead, we analyzed nuclear morphology and YAP localization in differentiating neuronal precursors, and found that a subpopulation of these neuronal precursors has flatter nuclear morphology and nuclear YAP localization, indicating that they are potentially mechanically more stressed. This effect is stronger in *Nup133*^{-/-} cells at multiple analyzed time-points after induction of neuronal differentiation, showing that it is likely not an effect of a potentially slightly offset neuronal differentiation speed (Fig. 7e-k, Extended Data Fig. 9e-i).

At last, we measured the nuclear envelope thickness, and compared it between mES and neural progenitor cells. We find that the NE in neural progenitor cells is significantly thinner than that in mES, in both wild-type and *Nup133*^{-/-} cells (Extended Data Fig. 2g-i). This would be consistent with the assumption that differentiation from mES into neural progenitor cells leads to increased tension on the nuclear envelope that pulls and stretches the membranes, thereby reducing the distance between inner and outer membrane. These data further support the idea that the mechanical load is increased in our neuronal progenitor samples compared to the respective mES cells, however, we note that we cannot directly measure membrane tension at this stage.

Referee #1 suggested using hypotonic shock to trigger nuclear deformation. As described in more detail below, we had attempted this experiment, yet did not observe conclusive effects.

Response to the reviewers' Comments:

Reviewer #1:

Remarks to the Author:

The manuscript by Taniguchi and co-workers analyze the structural changes of the nuclear pore upon ES cell differentiation into neural progenitors. While they observe no drastic changes, they find compositional changes and in particular a role for Nup133 in determining nuclear pore structure and integrity in neuronal progenitors. Deletion of Nup133 has been previously reported to lead to neuronal differentiation defects and the authors now report an important role in nuclear pore complex architecture and stability. Overall the manuscript describes intriguing and potentially important findings that will be interesting to the wider cell biology community. The manuscript is well written and the data largely of high quality.

A few issues should be addressed prior to publication, mainly related to the correlations between nuclear envelope tension, nuclear pore site fragility and DNA damage that appears somewhat underdeveloped and mostly correlative at this point.

1. The authors point to increased mechanical tension as the cause for nuclear pore fragility. However, it is not clear why this should be cell type specific. While the authors show pore dilation, this could be a result of tension-unrelated factors. Thus, the authors should address if NE tension or forces acting on the NE are in fact higher in neurons than in ES cells. Comparative quantitative analyses of NE tension/mechanics would greatly strengthen the manuscript.

As pointed out above, we analyzed nuclear morphology and YAP localization in neuronal precursors, and found that a subpopulation of these cells has a flatter nuclear morphology and nuclear YAP localization (Fig. 7e), indicating that they are potentially mechanically more stressed. This effect is stronger in *Nup133*^{-/-} cells at multiple analyzed time-points, showing that it is not an effect of a potentially slightly offset differentiation speed (Fig. 7e-k, Extended Data Fig. 9e-i).

Figure 7; (e) Representative immunofluorescence staining images of neuronal precursors with nuclear YAP at 48 h time point after neuronal differentiation induction. Maximum projection images are shown together with the X-Z orthogonal views of the nuclei highlighted with white arrowheads. Scale bars, 10

µm. (f - h) Mean percentage of nuclei with nuclear YAP localization in neuronal precursors at 2 h (f), 24 h (g), and 48 h (h) time points after neuronal differentiation induction. Data are from two (f) or four (g, h) biological replicates. Error bars denote standard deviations. Two-tailed Student's t-test; ** $p < 0.01$. (i - k) Mean percentage of nuclei with flatness index below 0.8 in neuronal precursors at 2 h (i), 24 h (j), and 48 h (k) time points after neuronal differentiation induction. Data are from two (i) or four (j, k) biological replicates. Error bars denote standard deviations. Two-tailed Student's t-test; * $p < 0.05$, ** $p < 0.01$.

Further, we measured the nuclear envelope thickness of wildtype mES and neural progenitor cells, and confirmed that NE in neural progenitor cells is significantly thinner than those in mES, in both wild-type and *Nup133*^{-/-} cells (Extended Data Fig. 2g-i). This observation is consistent with the notion of more stretched nuclear membranes.

Extended Data Figure 2; (g, h) Representative tomographic slices showing the side views of the nuclear envelope of the wild-type mES cells (g) and the wild-type neural progenitor cells (h). (i) Quantification of the nuclear envelope thickness. The thickness was measured manually at more than six representative positions within the tomograms, and the averaged values were plotted. Bars denote the means and standard deviations ($n = 15$ tomograms).

We have also edited the text to make it more clear that we cannot directly measure nuclear membrane tension at this stage.

2. The authors propose that increase in mechanical stress triggers the nuclear pore rupture. This should be demonstrated by triggering nuclear deformation by for example hypotonic shock

We had indeed attempted such experiments earlier using mES cells and did not obtain conclusive results. As pointed out above, we therefore attempted confinement experiments with mES cells during the revisions, which were also not conclusive. We think the issues in both cases are due to technical challenges, in particular the largely heterogenous morphology of the mES cells seeded as a monolayer without feeder cells. This would make them respond very differentially to mechanical stress, hampering us from accurately assessing the impact of experimental perturbation.

However, as pointed out above, we now included data on nuclear morphology and YAP localization in the neuronal precursors (Fig. 7e-k, Extended Data Fig. 9e-i), which are consistent with the proposed model of mechanical stress-triggered YAP nuclear import. Furthermore, we also included data on changed inner to outer nuclear membrane distance in the neural progenitors (Extended Data Fig. 2g-i), which was measured based on the tomographic data. All these results point to an increased mechanical load during all stages of differentiation, that in particular affects *Nup133*^{-/-} cells.

3. It is not clear why this nuclear pore fragility would cause widespread DNA damage. Is the DNA herniating into the cytoplasm? Is there micronucleus formation? Or could the effect be more indirect through regulation of transcription/replication conflicts for example that are known to be modulated through nuclear pore-related processes. The authors should analyse cGAS activation to assess DNA herniations as well as more carefully analyse nuclear shapes and micronuclear formation as well as address the role of replication stress, in particular as the analyses are done in S-phase synchronized cells.

Please also see the general response above. This is an interesting question that we admittedly cannot yet entirely clarify. Additional experiments during the revisions did not suggest micronuclei formation or cGAS activation in neuronal precursors. Also, in our tomograms of the neural progenitor cells, we observe gaps in the nuclear envelope, but no clear leakage of chromatin into the cytosol. We have meanwhile observed such a phenotype concomitantly with wider NE openings in another project that investigates aging (unpublished). We therefore think that in the present study, we are likely observing early events that in some, albeit rare, cases result in NE widening. But this conclusion remains speculative.

However, as pointed out above, additional analysis shows that 53BP1 foci accumulate in *Nup133*^{-/-} neuronal precursors together with γ -H2AX foci (Fig. 7a-c). These data support the idea that DNA damage and repair more often occurs in *Nup133*^{-/-} cells once they are differentiating under a likely more mechanically stressed condition that involves cell adhesion.

Figure 7; (a) Representative immunofluorescence staining images of neuronal precursors with γ -H2AX and 53BP1 foci. Maximum projection images were shown. Samples were prepared at 24 h time-point after induction of neuronal differentiation. (b, c) Mean percentage of nuclei with ≥ 5 γ -H2AX (b) and 53BP1 (c) foci in the wild-type and *Nup133*^{-/-} neuronal precursors.

Reviewer #2:

Remarks to the Author:

The manuscript by Taniguchi et al. examined NPC architecture in mouse embryonic stem cells (mES) and in neural progenitor cells in both wild-type and *Nup133*-deficient cells, using state-of-the-art cryo-electron tomography-based structural biology. Previous studies had shown that ESCs are unaffected without *Nup133*, while differentiated neurons are impacted. Additionally, prior research demonstrated that differentiation correlates with increased mechanical stress, and that NPCs can dilate upon mechanical stress. The current study provides a structural explanation for these observations by investigating the relationship between NPC diameter changes, cell differentiation, and disruption of NPC scaffold architecture. The hypothesis being that the NPC scaffold adapts to mechanical changes in the nuclear envelope during differentiation, with impaired responses in perturbed NPC architectures.

The study presents intriguing insights into NPCs lacking *Nup133*! I find it very exciting to see that in *Nup133*^{-/-} cells some 5% of NPCs have an altered rotational symmetry and, moreover, that a large fraction of NPCs have incomplete nuclear and cytoplasmic rings! These conclusions are drawn from three-dimensional template matching (TM). The incomplete presence of the CR and NR is proposed to weaken the ability of the NPC to adapt to forces stretching the nuclear envelope. Their incomplete presence correlates with the appearance of overstretched NPCs, with some NPCs losing connection to the pore membrane. These openings compromise nuclear compartmentalization, potentially leading to increased DNA damage. Overall, this is a very exciting and important structural study with – as far as I can judge – high quality data. The findings are important for the NPC community and researchers in developmental biology, neurobiology, and possibly cancer biology fields. The relationship to NE ruptures happening at NPCs is an exciting discussion point.

I enthusiastically support publishing this paper and only have a few minor suggestions for improving the manuscript.

We thank the reviewer for the positive assessment of our study.

Most importantly I feel that the phrasing in the text suggests at several places that the authors have data to support that the NPCs were fine at first, but that the NR and CRs disintegrate and detach as a result of mechanical stress (over-stretching) during differentiation. E.g., in line 316 they write “we conclude that an increase in nuclear envelope tension leads to NPC over-stretching, which can result in disintegration of NPCs” While that is not un-logical it is also not proven and it would be better to make that clear. If the NPCs disintegrate or even misassemble independent of stretching, all of their observations would still stand, e.g., their diameter would also increase.

We observe aberrant symmetries and incomplete CR and NR architectures (Fig. 3-5), as well as an overall increase in NPC diameter (Fig. 1d, 6c, d) in *Nup133*^{-/-} mES cells prior to inducing differentiation. However, the cases with more extreme diameter increase (over-stretching) are mainly observed in the *Nup133*^{-/-} neural progenitor cells (Fig. 6d), where we expect an increase in nuclear envelope tension.

As pointed out above, we cannot yet directly measure membrane tension in cells, but we obtained additional data supporting this model. In particular, we found that subpopulation of the neuronal precursors has flatter nuclear morphology and nuclear YAP localization. This effect is much stronger in *Nup133*^{-/-} cells (Fig. 7e-k, Extended Data Fig. 9e-i). Further, we

measured the nuclear envelope thickness and compared it between mES and neural progenitor cells. This confirmed that the NE in neural progenitor cells is significantly thinner than that in mES, in both wild-type and *Nup133*^{-/-} cells (Extended Data Fig. 2g-i). Both of these findings support the notion of an increased nuclear mechanical load upon differentiation induction.

As also discussed above, at this stage we cannot formally prove that an increase in membrane tension is the direct or sole cause for the observed changes. We have revised the text to make this more transparent.

*Together, these observations are consistent with the above-proposed model (Fig. 6a). We speculate that an increase in nuclear envelope tension arising from changes in nuclear morphology during differentiation can lead to NPC over-stretching and disintegration in the *Nup133*^{-/-} cells. Although the exact causal relationships, including formal proof for an increase in nuclear membrane tension, await further investigation, the frequent NPC disintegration observed in *Nup133*^{-/-} cells could well be a contributing factor to the previously described differentiation defect phenotype.*

This uncertainty also makes me somewhat less enthusiastic about the title which puts safeguarding nuclear integrity at the central place, while the data is most directly on NPC structural integrity.

We would prefer to keep the present title, as we think it best summarizes the broader implication of this work- the NPC is necessary to maintain an unperturbed nuclear envelope. With integrity of the nuclear envelope, we refer to intact membranes with NPCs. We clearly show that this is affected in *Nup133*^{-/-} neural progenitors, where NPC over-stretching and disintegration leads to large openings of the nuclear envelope. If NPCs cannot fulfill their safeguarding role this likely eventually affects nuclear envelope function. This title does not refer to any mechanistic details where the causal link is not yet entirely established, such as effects on genome integrity upon altered nuclear envelope tension.

Other minor comments:

1) The connection to DNA damage could be elucidated further, particularly regarding the link between decreased nuclear compartmentalization due to unstable *Nup113*-delta NPCs and increased γ -H2AX foci. Do the authors think there are more γ -H2AX foci because there is more damage or because the repair is impaired? How would either relate to NE ruptures or a compromised NPC permeability barrier? This is not so clear from the existing literature. Including the publications cited.

As discussed in detail in the general response, our data indicate that DNA repair pathways are not grossly impaired in *Nup133*^{-/-} cells, as we do not detect a clear difference in DNA damage sensitivity compared to wildtype cells (Extended Data Fig. 8g, h).

Extended Data Figure 8; (g, h) Dose-response viability curves showing the viability percentage of mES cells (g) and neuronal precursors (h) treated with different concentrations of DNA-damaging agents. The dots show the measured values from three independent experiments.

Further, we included additional analysis and show that 53BP1 foci accumulate in *Nup133*^{-/-} neuronal precursors together with γ -H2AX foci. These data support the conclusion that DNA damage and repair more often occurs in *Nup133*^{-/-} cells when they undergo differentiation (Fig. 7a-c).

2) Figure 1A,B, S3, Figure 2: it would be useful to point out the protrusion from the NR at the Nup107/Nup133 region, which the authors write “likely corresponding to nuclear basket filaments.” It is exciting in itself that these filaments are resolved and also that this protrusion is less clear in the *nup133*^{-/-}.

We have added additional labels to the figures to clarify where the basket filament is. Prior work has indeed shown that basket assembly is perturbed in the *Nup133*^{-/-} mES cells (Souquet *et al. Cell Reports*, 2018).

3) Regarding the detachment of the CRs reported in (Fig. 6B, S6A), some images may not clearly show this detachment, such as #13 and #15.

We thank the reviewer for this comment. Indeed, the detachment is less drastic in #13 and #15 compared to the others when visualized from the top. For clarity, we have added further labels to the middle images of the figures in Extended Data Fig. 7 and Extended Data Fig. 10. We also added additional labels to Fig.6b and Fig.8a, b to clarify the figures.

4) P 7: Consider rephrasing “Despite this critical scaffolding role and its clear presence within the averages of the wild-type NPC (Fig.136 1B), *Nup133* is dispensable for the proliferation of mES cells”, removing the technical interruption regarding its clear presence within wild-type NPC averages.

As suggested, we have changed that sentence to: “*Despite this critical scaffolding role, Nup133 is dispensable for the proliferation of mES cells*”

5) Does the new structural data provide an explanation why Nup133 deletion results in clustering of NPCs in yeast?

NPC clustering was not observed in Souquet *et al.*, *Cell Reports*, 2018 in the *Nup133^{-/-}* mES cells. Thus, clustering could be a yeast-specific phenotype, which may not be related to the phenotypes observed here. To clarify this, the question would need to be directly addressed in yeast experiments. Based on our structural data from mouse ES cells, we would rather not speculate.

Reviewer #3:

Remarks to the Author:

Nuclear pore complexes (NPCs) regulate exchange of molecules between the nuclear and cytoplasmic environments. NPCs are also an integral part of the overall architecture of the nucleus, which houses the genetic contents of cells. Mutations in NPC subunit proteins (nucleoporins or Nups) contribute to many diseases, but the mechanisms of action are unclear. Taniguchi and colleagues investigate the deletion of Nup133 in mouse embryonic stem cells and use this as a system to investigate how Nup133 deletion alters NPC architecture during differentiation. I find this study to be fantastic, and a foundational building block for the field to understand how disease mutations alter cellular differentiation via introducing a plethora of architectural variability within NPCs. Moreover, the authors use their observations to generate key hypotheses on how fundamental the architectural plasticity of NPCs is for development. Furthermore, the structural work performed here is world class. This paper is absolutely fit for Nature Cell Biology. I congratulate the authors on these findings, which are broadly interesting to those in the NPC field, cellular development, disease pathologies, and cryo-ET. I include comments below.

1. Fig. S1D - an example raw image of cells stained with the Pax6 antibody should be shown prior to and after differentiation for WT and Nup133 deletion.

As suggested, we now include the Pax6 and Oct4 immunofluorescence in Extended Data Fig. 1e-h. As expected, these data confirm successful differentiation.

Extended Data Figure 1; (e, f) Representative immunofluorescence staining images of Pax6 in mES cells (e) and neural progenitor cells shortly after neuronal differentiation induction (f). (g, h) Representative immunofluorescence staining images of Oct3/4 in mES cells (g) and differentiating neural progenitor cells shortly after neuronal differentiation induction (h). In (e - h), single image frames from confocal stacks are shown.

2. Fig 1C - is it possible for the authors to measure the curvature change between mES and differentiated progenitor?

As suggested, we measured the angle for individual subtomograms, based on the positions of IR and membrane attachment sites of Y complexes. The results show that the angle is wider in the WT neural progenitor NPCs than in WT mES NPCs, while no obvious difference between *Nup133^{-/-}* mES and neural progenitor NPCs is detectable. This data is in line with structural data showing that the inner ring moves more during NPC dilation, while the cytoplasmic and nuclear rings are more rigid (Mosalaganti *et al*, *Science*, 2022). This data is now included in Extended Data Fig. 3.

Extended Data Figure 3; (f) Schematic representation showing the workflow of angle measurement. Based on the subtomogram averages of individual rings (CR, IR, and NR), five coordinates were derived for each NPC subunits (depicted as green spheres). Subsequently, the angle between two planes (one formed between IR and two CR coordinates, the other formed between IR and two NR coordinates) were measured for each NPC subunits. (g) Results of the angle measurement. The angles measured for individual subunits were averaged per NPC, and the averaged angles were plotted. Bars denote the means and standard deviations (n = 184 NPCs for the wild-type mES dataset, n = 119 NPCs for the wild-type neural progenitor dataset, n = 128 NPCs for the *Nup133*^{-/-} mES dataset, n = 136 NPCs for the *Nup133*^{-/-} neural progenitor dataset).

3. Fig 1B - the *Nup133* region of the map would be more clear if the authors colored the zone of the map.

As suggested, we have changed the figure.

4. Page 9, line 185 - should Figure 2 here actually refer to Figure 3?

The reviewer is most likely referring to this sentence: “*Visual inspection of the reconstructed tomograms revealed that NPCs with non-canonical 7- or 9-fold symmetric architectures are present in the tomograms from the *Nup133*^{-/-} mES cells, in addition to the canonical 8-fold symmetric NPCs (Fig. 2A-F).*”. As the reviewer pointed out, Fig. 3a-f should be referred to here. It is corrected in the revised manuscript.

5. Figure 3 - This figure would be strengthened by plotting the overall populations of each symmetry group here to highlight the differences between WT and *Nup133* del.

We have included a pie chart in the new Figure 3 to make this more obvious.

Figure 3; (g) Pie chart showing the distribution of particles after reference-based classification. Particles with ambiguous class assignment were indicated with light grey, while particles discarded before classification due were indicated with dark grey.

6. In general, I find the high resolution template matching to be quite powerful in showing the degree of heterogeneity in the *Nup133* del in each layer of the NPCs.

We agree with the reviewer that this approach is powerful.

7. Fig 7 - the authors should compare the change in H2AX foci between mES and differentiated progenitor to strengthen the point. It is important to know the extent of DNA damage prior to differentiation in this context.

As suggested, we have now also assessed γ -H2AX foci and 53BP1 foci in mES cells. As discussed in the general response, we do not observe a striking difference between wildtype and *Nup133*^{-/-} mES cells. This data is now included in Extended Data Fig. 8a-c. This clarifies that the increased accumulation of DNA damage in the *Nup133*^{-/-} cells in comparison to the wildtype cells is observed specifically in neuronal precursors undergoing neuronal differentiation.

We however do not believe that it is possible to directly compare the extent of DNA damage in mES and neural progenitor cells using these markers. Constitutive phosphorylation of H2AX has previously been shown in mES cells in culture and during early mouse development. In mouse preimplantation development and embryos, γ -H2AX was reported to be detectable in absence of DNA damage, and did not colocalize with 53BP1 that was detectable in certain stages of development (Ziegler-Birling *et al. Int. J. Dev Biol.*, 2009). A more recent study confirmed this and showed that γ -H2AX foci form in unperturbed mES cells and found that they are linked to replication stress. Importantly, the authors show that γ -H2AX as well as markers of single-stranded DNA and replication fork reversal are rapidly suppressed upon induction of differentiation (Ahuja *et al., Nat. Commun.*, 2016). We thus assume that the γ -H2AX foci and 53BP1 foci we observe in mES cells are resolved when

cells differentiate into neural progenitors and that the events we observe in neuronal precursors are indeed events of DNA damage incurred later on.

Extended Data Figure 8; (a) Representative immunofluorescence staining images of mES cells with γ -H2AX and 53BP1 foci. Maximum projection images of five central confocal frames are shown. Scale bars, 10 μ m. (b, c) Mean percentage of nuclei with ≥ 5 γ -H2AX (b) and 53BP1 (c) foci in the wild-type and *Nup133*^{-/-} mES cells. Data are from three biological replicates. Error bars denote standard deviations.

To further address the reviewer's comment, we have also additionally assessed γ -H2AX foci and 53BP1 foci in neuronal precursors at 2 h time point after neuronal differentiation induction. Unlike the samples from 24 h time point (Fig. 7a-c), we do not observe a significant difference between wildtype and *Nup133*^{-/-} at this earlier time point (Extended Data Fig. 8d-f). This data further supports our notion that the increased accumulation of DNA damage in the *Nup133*^{-/-} cells in comparison to the wildtype cells occurs during neuronal differentiation.

Extended Data Figure 8; (d) Representative immunofluorescence staining images of neuronal precursors 2 h after neuronal differentiation induction with γ -H2AX and 53BP1 foci. Maximum projection images are shown. Scale bars, 10 μ m. (e, f) Mean percentage of nuclei with ≥ 5 γ -H2AX (e) and 53BP1 (f) foci in the wild-type and *Nup133*^{-/-} neuronal precursors 2 h after neuronal differentiation induction. Data are from three biological replicates. Error bars denote standard deviations.

8. It would be interesting for the authors to speculate more in the discussion on the functional consequences of the symmetry mismatches they observe via TM in CRs and NRs. Is the functional consequence of this only envelope integrity? How would this affect transport across the NPC?

It is difficult to predict how transport would be affected by the change in symmetry, as we do not know if transport receptor binding is altered, in particular as NPCs in *Nup133*^{-/-} cells likely lack a nuclear basket structure. Also, NPCs with non-canonical symmetry are roughly 15% of the entire population, so it is hard to conceive that those small populations of NPCs would generally change nuclear transport activity. Since the sizes of their central channels differ from that of 8-fold symmetric NPCs (see Fig. 3a-f), it may change the nuclear envelope permeability. Yet, we currently do not have any supportive evidence. Thus, we would rather not speculate in our current manuscript.

Point-by-point response

Reviewer #1 (Remarks to the Author):

The authors have made substantial efforts to address concerns of this reviewer. While some aspects (source of DNA damage, nuclear envelope tension) remain partially unanswered, these are challenging questions that require additional in-depth studies and I appreciate that authors transparency in expressing limitations of their approaches. The manuscript is in good shape for publication.

We thank the reviewer for appreciating our revision and the positive assessment of the manuscript.

Reviewer #2 (Remarks to the Author):

The additional cell biological data on 53BP1 foci, CGAs, BAF and YAP localization, flatness and sensitivity to DNA-damaging agents further strengthened the conclusions of the work. The textual edits take care of resolving some unclarities. I enthusiastically support publication of the work.

a minor suggestion would be to indicate in figure 7 and extended data 9 which cells are considered to have less than 5 foci, as just judging by eye, the images may not be representative of the quantification.

We thank the reviewer for the helpful comments throughout. We have now revised Figure 7a, Extended Data Figure 8a and 8d. To clarify how foci-positive nuclei look, we have replaced the panels with ones in which examples of nuclei with more than 5 detected foci are indicated with arrowheads and highlighted in the insets.

Reviewer #3 (Remarks to the Author):

The authors addressed my concerns, I support publication.

We thank the reviewer for the positive assessment of our manuscript.